# Variational Supervised Contrastive Learning

**Ziwen Wang**[*]
University of Illinois
Urbana-Champaign
ziwen2@illinois.edu

**Jiajun Fan**
University of Illinois
Urbana-Champaign
jiajunf3@illinois.edu

**Thao Nguyen**
University of Illinois
Urbana-Champaign
thaotn2@illinois.edu

**Heng Ji**
University of Illinois
Urbana-Champaign
hengji@illinois.edu

**Ge Liu**
University of Illinois
Urbana-Champaign
geliu@illinois.edu

## Abstract

Contrastive learning has proven to be highly efficient and adaptable in shaping representation spaces across diverse modalities by pulling similar samples together and pushing dissimilar ones apart. However, two key limitations persist: (1) Without explicit regulation of the embedding distribution, semantically related instances can inadvertently be pushed apart unless complementary signals guide pair selection, and (2) excessive reliance on large in-batch negatives and tailored augmentations hinders generalization. To address these limitations, we propose Variational Supervised Contrastive Learning (VarCon), which reformulates supervised contrastive learning as variational inference over latent class variables and maximizes a posterior-weighted evidence lower bound (ELBO) that replaces exhaustive pair-wise comparisons for efficient class-aware matching and grants fine-grained control over intra-class dispersion in the embedding space. Trained exclusively on image data, our experiments on CIFAR-10, CIFAR-100, ImageNet-100, and ImageNet-1K show that VarCon (1) achieves state-of-the-art performance for contrastive learning frameworks, reaching 79.36% Top-1 accuracy on ImageNet-1K and 78.29% on CIFAR-100 with a ResNet-50 encoder while converging in just 200 epochs; (2) yields substantially clearer decision boundaries and semantic organization in the embedding space, as evidenced by KNN classification, hierarchical clustering results, and transfer-learning assessments; and (3) demonstrates superior performance in few-shot learning than supervised baseline and superior robustness across various augmentation strategies. Our code is available at https://github.com/ziwenwang28/VarContrast.

## 1 Introduction

Ever since its introduction, contrastive learning has become a central paradigm in representation learning, enabling advances across computer vision, natural language processing (NLP), speech, multimodal understanding, and applications in natural sciences [62, 65, 24, 64, 3, 41, 35]. Foundational models such as SimCLR [10], MoCo [31], BYOL [29], and the fully supervised SupCon [37] have enabled capabilities ranging from zero-shot image classification to state-of-the-art sentence embeddings, demonstrating that the simple "pull-together, push-apart" principle scales effectively across domains. Despite their empirical success, these objectives function as heuristic energy functions with opaque statistical meaning. While recent analyses have made progress [32, 46, 53, 47, 61], we still

---

[*]Corresponding author.

39th Conference on Neural Information Processing Systems (NeurIPS 2025).

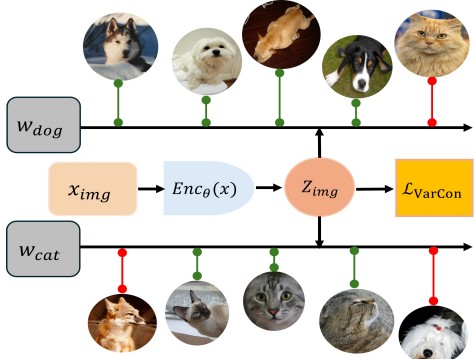
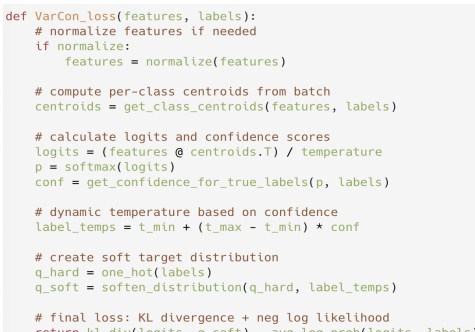

```python
def VarCon_loss(features, labels):
    # normalize features if needed
    if normalize:
        features = normalize(features)

    # compute per-class centroids from batch
    centroids = get_class_centroids(features, labels)

    # calculate logits and confidence scores
    logits = (features @ centroids.T) / temperature
    p = softmax(logits)
    conf = get_confidence_for_true_labels(p, labels)

    # dynamic temperature based on confidence
    label_temps = t_min + (t_max - t_min) * conf

    # create soft target distribution
    q_hard = one_hot(labels)
    q_soft = soften_distribution(q_hard, label_temps)

    # final loss: KL divergence + neg log likelihood
    return kl_div(logits, q_soft) - avg_log_prob(logits, labels)
```

Figure 1: VarCon architectural flowchart and Pseudocode. **Left:** Input images are processed through an encoder network to produce $\ell_2$-normalized embeddings $\boldsymbol{z}$. Class-level centroids $\boldsymbol{w}_r$ are computed dynamically from mini-batch embeddings. The model determines sample's classification difficulty and applies confidence-adaptive temperature scaling $\tau_2(\boldsymbol{z})$, which tightens constraints on challenging samples and relaxes them for well-classified examples. **Right:** Pseudocode implementation of our ELBO-derived loss function combining KL divergence and negative log-likelihood terms.

lack a principled account of how contrastive interactions shape embeddings that capture relational structure among data samples. This gap motivates our work to establish a rigorous foundation for contrastive learning through variational inference.

Generative models provide a complementary perspective: they introduce latent variables and estimate parameters via maximum likelihood, assigning explicit probabilistic semantics to the representation space [51]. In Variational Autoencoders (VAEs) [38] and flow-based architectures [42, 12, 22, 60], Euclidean proximity in the latent space corresponds to regions of high data likelihood, endowing distances with a rigorous statistical interpretation. Although contrastive learning is not inherently likelihood-based, recent theory frames InfoNCE as density-ratio maximization: it optimizes the ratio of joint probability over marginal probabilities for positive pairs relative to negatives, prioritizing genuine pairs. Both paradigms ultimately seek embeddings where "near" implies "probable," though via different routes—generative models integrate over latent variables to maximize likelihood, while contrastive methods directly sculpt the space through positive-negative comparisons. This conceptual convergence invites our unified framework, where variational inference provides probabilistic grounding for contrastive objectives while preserving their proven geometric inductive biases.

In this work, we present Variational Supervised Contrastive Learning (VarCon), a probabilistically grounded framework that preserves the geometric structure of conventional contrastive objectives while endowing them with explicit likelihood semantics. VarCon treats the class label as a latent variable and maximizes a posterior-weighted evidence lower bound (ELBO), replacing hard labels with soft probabilities reflecting the model's current belief. This formulation confers three key benefits. First, each embedding interacts with a single class-level direction computed on the fly, reducing computation from quadratic to nearly linear in batch size. Second, it employs a confidence-adaptive temperature that tightens pull strength for hard samples and relaxes it for confident ones, providing fine-grained control over intra-class compactness. Third, we develop a novel objective with two synergistic terms: a Kullback–Leibler divergence aligning the auxiliary posterior with the model's class posterior, and a negative log-likelihood term for the ground-truth label. This approach simultaneously aligns distributions, maximizes class likelihood, and prevents representational collapse. Our main contributions are summarized as follows:

1. We propose a novel formulation of contrastive learning in variational inference with an explicit ELBO framework, deriving the VarCon loss that explicitly regulates embedding distributions and enforces appropriate semantic relationships between samples.

2. We propose a confidence-adaptive temperature scaling for label softening strategy that pushes the edge of learning hard positives and negatives, proven through gradient derivation analyses.

3. We advance state-of-the-art contrastive learning performance across multiple architectures, achieving 79.36% Top-1 accuracy on ImageNet with ResNet-50 (vs. 78.72% for SupCon) and 81.87% with

ResNet-200, while converging in 200 epochs versus SupCon's 350 and using smaller batch sizes (2048 vs. 4096).

4. Our learned representations demonstrate superior semantic organization with clearer hierarchical clustering, and maintain strong performance in low-data settings and various augmentation strength.

## 2 Preliminaries

**Contrastive Learning.** Early metric-learning objectives, the pairwise contrastive loss [13], triplet margin loss [54], and N-pair loss [56], laid the groundwork for modern contrastive frameworks. In contrastive learning, given an input sample $x$ with ground-truth class label $r$, an encoder maps $x$ to an embedding $z \in \mathbb{R}^d$, typically $\ell_2$-normalized. The field subsequently converged on noise-contrastive formulations like InfoNCE [46]:

$$\mathcal{L}_{\text{InfoNCE}} = -\log \frac{\exp(z_i^\top z_j^+/\tau)}{\sum_{k=1}^{N} \mathbf{1}_{[k \neq i]} \exp(z_i^\top z_k/\tau)}$$

where $z_i$ and $z_j^+$ are positive pairs and $\tau$ is a temperature parameter. InfoNCE drives self-supervised systems like SimCLR [10] and MoCo [11, 31], and scales to cross-modal setups in CLIP [48]. For supervised learning, SupCon [37] extends this by treating same-class samples as positives:

$$\mathcal{L}_{\text{SupCon}} = \sum_{i \in I} \frac{-1}{|P(i)|} \sum_{p \in P(i)} \log \frac{\exp(z_i^\top z_p/\tau)}{\sum_{a \in A(i)} \exp(z_i^\top z_a/\tau)}$$

where $P(i)$ contains samples of the same class as $i$, and $A(i)$ includes all samples except $i$. Recent theoretical advances [68, 34] and methods like X-CLR [55] further improve these frameworks by analyzing optimization dynamics and introducing graded similarities. Our approach builds on these foundations by reformulating supervised contrastive learning through variational inference while preserving semantic linkages across classes. Additional related work is provided in Appendix E.

**Variational Inference.** Variational methods provide principled probabilistic frameworks for learning latent representations by maximizing an evidence lower bound (ELBO). Variational Autoencoders (VAEs) [38, 50] pioneered this approach with amortized encoder-decoder architectures learning continuous latent spaces. Later advances improved flexibility through normalizing flows [39, 49] and learned priors, while Bayesian approaches like Bayes-by-Backprop [7] and Monte-Carlo dropout [23] incorporated parameter uncertainty. Recent work has integrated contrastive learning with variational objectives: Noise-Contrastive Prior [2] separates between posteriors and priors to tighten latent fit; variational information-bottleneck augments SimCLR [59]; and Recognition-Parameterised Models reformulate the ELBO to yield InfoNCE-style objectives [1]. These integrations create latent spaces that are simultaneously generative, predictive, and uncertainty-aware. Our approach builds on these foundations by maximizing a class-conditional data likelihood with label-softened targets, preserving contrastive learning's geometric properties while incorporating explicit probabilistic semantics.

**Soft Labeling.** Soft labeling replaces hard one-hot targets with distribution-based supervision, first popularized through label-smoothing in Inception-v3 [57] and knowledge distillation [33]. Recent advances employ soft labels to improve representation learning [21, 20]: for noisy targets, smoothed distributions enhance calibration across facial expressions [43] and enlarge inter-class margins in partial-label settings [28]. This concept extends to semi-supervised learning where ESL [44] treats uncertain pixels as soft pseudo-labels, SoftMatch [9] applies confidence-weighted pseudo-labels, and ProtoCon [45] refines labels through online clustering. For cross-modal learning, SoftCLIP [25] strengthens image-text embeddings by weighting related captions. Our approach employs soft labeling as a confidence-adaptive variational distribution that dynamically adjusts class distributions, more uniform for confident samples and more peaked for challenging ones, to enhance representation learning through targeted supervision.

## 3 Method

Throughout this study, we use the following notation: Let $z \in \mathbb{R}^d$ denote the $\ell_2$-normalized embedding produced by the encoder (e.g., ResNet-50) for an input sample $x$ with ground-truth class index $r \in \{1, \ldots, C\}$. Each class is represented by a unit-norm reference vector $w_r \in \mathbb{R}^d$,

obtained by normalizing the class mean: $\boldsymbol{w}_r = \bar{\boldsymbol{z}}_r / \|\bar{\boldsymbol{z}}_r\|_2$ where $\bar{\boldsymbol{z}}_r = |\mathcal{B}_r|^{-1} \sum_{i \in \mathcal{B}_r} \boldsymbol{z}_i$ and $\mathcal{B}_r = \{\, i \mid r_i = r, \, i \in \mathcal{B} \,\}$ denotes the index set of samples from class $r$ in mini-batch $\mathcal{B}$. We set $\tau_1$ to be the fixed temperature that scales the logits to adjust the sharpness of the class distribution and $\tau_2(\boldsymbol{z})$ the confidence-adaptive temperature that softens each sample's target distribution to regularize learning and control intra-class dispersion. With $\theta$ being the set of learnable weights of the encoder, $p_\theta(r \mid \boldsymbol{z})$ denotes probability the model assigns to the correct class $r$ for embedding $\boldsymbol{z}$, $p_\theta(\boldsymbol{z} \mid r)$ the class-conditional likelihood of observing $\boldsymbol{z}$, and $p_\theta(\boldsymbol{z}) = \sum_{r'} p_\theta(\boldsymbol{z} \mid r')p(r')$, with $r'$ the dummy class index, gives the marginal embedding density used in the variational derivation. We introduce $q_\phi(r \mid \boldsymbol{z})$ as a confidence-adaptive label distribution parameterized by $\phi$, which plays the role of an adaptive temperature for accentuating the probability assigned to the ground-truth class while allocating the remaining mass uniformly across the other $C - 1$ classes.

## 3.1  The Variational Bound of Class-Conditional Likelihood

To formulate our variational approach, we need to establish an evidence lower bound (ELBO) for the class-conditional likelihood of embeddings. We begin with Bayes' rule [26, 6], which allows us to express the class-conditional likelihood in terms of the posterior probability and marginal densities:

$$p_\theta(\boldsymbol{z} \mid r) \;=\; \frac{p_\theta(r \mid \boldsymbol{z})\, p_\theta(\boldsymbol{z})}{p(r)}. \tag{1}$$

Taking logarithms on both sides gives us:

$$\log p_\theta(\boldsymbol{z} \mid r) \;=\; \log p_\theta(r \mid \boldsymbol{z}) \;+\; \log p_\theta(\boldsymbol{z}) \;-\; \log p(r). \tag{2}$$

The marginal embedding density $p_\theta(\boldsymbol{z})$ integrates over all possible class assignments. By introducing our auxiliary distribution $q_\phi(r' \mid \boldsymbol{z})$ that sums to 1, and applying the identity $p_\theta(\boldsymbol{z}) = \sum_{r'} p_\theta(\boldsymbol{z} \mid r')p(r')$, we can rewrite the marginal density as:

$$p_\theta(\boldsymbol{z}) = \sum_{r'=1}^{C} \left[ p_\theta(\boldsymbol{z} \mid r')\, p(r')\, \frac{q_\phi(r' \mid \boldsymbol{z})}{q_\phi(r' \mid \boldsymbol{z})} \right]. \tag{3}$$

Substituting Eq. (3) into Eq. (2), we obtain:

$$
\begin{aligned}
\log p_\theta(\boldsymbol{z} \mid r) &= \log p_\theta(r \mid \boldsymbol{z}) \;+\; \log p_\theta(\boldsymbol{z}) \;-\; \log p(r) \\
&= \log p_\theta(r \mid \boldsymbol{z}) \;+\; \log \sum_{r'=1}^{C} \left[ p_\theta(\boldsymbol{z} \mid r')\, p(r')\, \frac{q_\phi(r' \mid \boldsymbol{z})}{q_\phi(r' \mid \boldsymbol{z})} \right] \;-\; \log p(r) \\
&= \log p_\theta(r \mid \boldsymbol{z}) \;+\; \log \sum_{r'=1}^{C} \left[ q_\phi(r' \mid \boldsymbol{z})\, \frac{p_\theta(\boldsymbol{z} \mid r')\, p(r')}{q_\phi(r' \mid \boldsymbol{z})} \right] \;-\; \log p(r).
\end{aligned}
\tag{4}
$$

From Eq. (1), we know that $p_\theta(\boldsymbol{z} \mid r)\, p(r) = p_\theta(r \mid \boldsymbol{z})\, p_\theta(\boldsymbol{z})$. Using this relation to substitute for $p_\theta(\boldsymbol{z} \mid r')\, p(r')$ in Eq. (4), and applying Jensen's inequality to the logarithm of a sum (since log is a concave function), we derive the ELBO for $\log p_\theta(\boldsymbol{z} \mid r)$:

$$\log p_\theta(\boldsymbol{z} \mid r) = \log p_\theta(r \mid \boldsymbol{z}) + \log \sum_{r'=1}^{C} \left[ q_\phi(r' \mid \boldsymbol{z}) \frac{p_\theta(\boldsymbol{z} \mid r') \, p(r')}{q_\phi(r' \mid \boldsymbol{z})} \right] - \log p(r)$$

$$= \log p_\theta(r \mid \boldsymbol{z}) + \log \sum_{r'=1}^{C} \left[ q_\phi(r' \mid \boldsymbol{z}) \frac{p_\theta(r' \mid \boldsymbol{z}) \, p_\theta(\boldsymbol{z})}{q_\phi(r' \mid \boldsymbol{z})} \right] - \log p(r)$$

$$\geq \log p_\theta(r \mid \boldsymbol{z}) + \sum_{r'=1}^{C} q_\phi(r' \mid \boldsymbol{z}) \log \left[ \frac{p_\theta(r' \mid \boldsymbol{z}) \, p_\theta(\boldsymbol{z})}{q_\phi(r' \mid \boldsymbol{z})} \right] - \log p(r)$$

$$= \log p_\theta(r \mid \boldsymbol{z}) + \sum_{r'=1}^{C} q_\phi(r' \mid \boldsymbol{z}) \left[ \log p_\theta(r' \mid \boldsymbol{z}) + \log p_\theta(\boldsymbol{z}) - \log q_\phi(r' \mid \boldsymbol{z}) \right] - \log p(r)$$

$$= \log p_\theta(r \mid \boldsymbol{z}) + \sum_{r'=1}^{C} q_\phi(r' \mid \boldsymbol{z}) \log p_\theta(\boldsymbol{z})$$

$$\hspace{3cm} + \sum_{r'=1}^{C} q_\phi(r' \mid \boldsymbol{z}) \left[ \log p_\theta(r' \mid \boldsymbol{z}) - \log q_\phi(r' \mid \boldsymbol{z}) \right] - \log p(r) \tag{5}$$

$$= \log p_\theta(r \mid \boldsymbol{z}) + \log p_\theta(\boldsymbol{z}) \sum_{r'=1}^{C} q_\phi(r' \mid \boldsymbol{z}) - D_{\mathrm{KL}}\big(q_\phi(r' \mid \boldsymbol{z}) \, || \, p_\theta(r' \mid \boldsymbol{z})\big) - \log p(r)$$

$$= \log p_\theta(r \mid \boldsymbol{z}) + \log p_\theta(\boldsymbol{z}) - D_{\mathrm{KL}}\big(q_\phi(r' \mid \boldsymbol{z}) \, || \, p_\theta(r' \mid \boldsymbol{z})\big) - \log p(r).$$

This derived bound consists of several interpretable components: (1) a term encouraging the model to correctly classify the input embedding, (2) the log marginal probability of the embedding, (3) a KL divergence that aligns our auxiliary distribution with the model's class posterior, and (4) a constant class prior. This ELBO serves as the foundation for our variational contrastive learning objective.

## 3.2 Variational Contrastive Learning

Having derived the ELBO for $\log p_\theta(\boldsymbol{z} \mid r)$ in Eq. (5), we observe that under contrastive learning settings, certain terms are either intractable or uninformative: $\log p_\theta(\boldsymbol{z})$ would encourage high likelihood throughout the embedding space without class distinction, while $\log p(r)$ is a fixed constant based on the dataset's class distribution. Therefore, instead of directly maximizing the full ELBO:

$$\mathcal{L}_{\mathrm{ELBO}} = D_{\mathrm{KL}}(q_\phi(r' \mid \boldsymbol{z}) \, || \, p_\theta(r' \mid \boldsymbol{z})) + \log p(r) - \log p_\theta(r \mid \boldsymbol{z}) - \log p_\theta(\boldsymbol{z}), \tag{6}$$

we focus on minimizing the contrastive-relevant components:

$$\mathcal{L}_{\mathrm{VarCon}} = D_{\mathrm{KL}}(q_\phi(r' \mid \boldsymbol{z}) \, || \, p_\theta(r' \mid \boldsymbol{z})) - \log p_\theta(r \mid \boldsymbol{z}). \tag{7}$$

This formulation balances two complementary objectives: the KL divergence term aligns our auxiliary distribution with the model's predictive distribution, while the log-posterior term encourages correct class assignments. To compute class probabilities efficiently, we leverage class centroids rather than pairwise comparisons. For each class $r$, we calculate a reference vector $\boldsymbol{w}_r = |\mathcal{B}_r|^{-1} \sum_{i \in \mathcal{B}_r} \boldsymbol{z}_i$, where $\mathcal{B}_r = \{\, i \mid r_i = r,\ i \in \mathcal{B} \,\}$ contains indices of samples with class $r$ in batch $\mathcal{B}$. The posterior probability is then:

$$p_\theta(r \mid \boldsymbol{z}) = \frac{\exp(\boldsymbol{z}^\top \boldsymbol{w}_r / \tau_1)}{\sum_{r'} \exp(\boldsymbol{z}^\top \boldsymbol{w}_{r'} / \tau_1)}, \tag{8}$$

with logarithm:

$$\log p_\theta(r \mid \boldsymbol{z}) = \log \left[ \frac{\exp(\boldsymbol{z}^\top \boldsymbol{w}_r / \tau_1)}{\sum_{r'} \exp(\boldsymbol{z}^\top \boldsymbol{w}_{r'} / \tau_1)} \right] = \frac{\boldsymbol{z}^\top \boldsymbol{w}_r}{\tau_1} - \log \sum_{r'} \exp \left( \frac{\boldsymbol{z}^\top \boldsymbol{w}_{r'}}{\tau_1} \right), \tag{9}$$

where $\tau_1$ is a fixed temperature parameter. For the target distribution $q_\phi$, we start with a one-hot distribution:

$$q_{\mathrm{one\text{-}hot}}(r' \mid \boldsymbol{z}) = \begin{cases} 1, & \text{if } r' = r, \\ 0, & \text{otherwise.} \end{cases} \tag{10}$$

We then apply adaptive softening using temperature $\tau_2$:

$$q_{\text{exp}}(r \mid \boldsymbol{z}) = 1 + [\exp(1/\tau_2) - 1] \, q_{\text{one-hot}}(r \mid \boldsymbol{z}), \tag{11}$$

and normalize to obtain the final distribution:

$$q_\phi(r \mid \boldsymbol{z}) = \frac{q_{\text{exp}}(r \mid \boldsymbol{z})}{\sum_{r'} q_{\text{exp}}(r' \mid \boldsymbol{z})} \tag{12}$$

A key innovation in our approach is using a confidence-adaptive temperature $\tau_2$ that varies between bounds $\tau_1 - \epsilon$ and $\tau_1 + \epsilon$:

$$\tau_2 = (\tau_1 - \epsilon) + 2 \, \epsilon \, p_\theta(r \mid \boldsymbol{z}), \tag{13}$$

where $\epsilon$ is a learnable parameter. When $p_\theta(r \mid \boldsymbol{z})$ is high (confident prediction), $\tau_2$ approaches $\tau_1 + \epsilon$, making $q_\phi(r \mid \boldsymbol{z})$ more uniform. Conversely, for difficult samples with low $p_\theta(r \mid \boldsymbol{z})$, $\tau_2$ approaches $\tau_1 - \epsilon$, creating a sharper distribution. This adaptive mechanism dynamically adjusts supervision intensity—relaxing constraints on well-classified samples while focusing learning on challenging ones (see gradient derivation in C.1). By minimizing $\mathcal{L}_{\text{VarCon}}$, we simultaneously control intra-class dispersion through the KL term and enhance the distinguishing power of embeddings through the posterior term, creating a representation space with clear semantic structure and decision boundaries.

## 4   Experiments

We evaluate our VarCon loss on four standard benchmarks: CIFAR-10, CIFAR-100 [40], ImageNet-100, and ImageNet [52, 18], using the official test splits. Our experiments demonstrate rapid convergence under various training configurations, including different batch sizes, epochs, and hyperparameter settings. For downstream classification, we freeze the pretrained encoder and train only a linear classification layer, and additionally apply KNN-classifier to the embeddings to investigate the learned semantic structure. To isolate the contribution of our learning objective, we employ the same data augmentation techniques and encoder architectures used in previous contrastive learning models [37]: augmentation strategies including SimAugment, AutoAugment [14], and StackedRandAugment [15] with ResNet-50 [30], ResNet-101, ResNet-200, and ViT-Base [19] architectures. We set the learning rate according to the linear-scaling rule lr $\propto B/256$ with cosine decay. Our best results are achieved with batch sizes of 512 for CIFAR-10/100 (200 epochs), 1,024 for ImageNet-100 (200 epochs), and 4,096 for ImageNet (350 epochs). Throughout, we use SGD with momentum 0.9 and weight decay $10^{-4}$ for smaller datasets including CIFAR-10, CIFAR-100, and ImageNet-100 and LARS [63] optimizer for ImageNet training to ensure stability. Full hyperparameter settings and optimization details are available in Appendix B.1. Analyses of the training dynamics, such as loss convergence and memory efficiency, are provided in Appendix D.

### 4.1   Classification Performance

To evaluate our proposed VarCon, we conducted extensive experiments on multiple benchmark datasets for image classification. Table 1 presents performance comparisons against state-of-the-art self-supervised methods (SimCLR [10], MoCo V2 [31, 11], BYOL [29], SwAV [8], VicReg [5], and Barlow Twins [66]) and supervised approaches (standard Cross-Entropy and SupCon [37]). VarCon consistently outperforms all competing methods across all datasets. Compared to SupCon, VarCon achieves 0.43% higher Top-1 accuracy on CIFAR-10 (95.94% vs. 95.51%), 1.72% higher on CIFAR-100 (78.29% vs. 76.57%), 1.28% higher on ImageNet-100 (86.34% vs. 85.06%), and 0.64% higher on ImageNet (79.36% vs. 78.72%). The advantage is more pronounced when compared to self-supervised methods. On ImageNet-100, VarCon surpasses the best self-supervised method (Barlow Twins at 80.83%) by 5.51%, and on ImageNet, it outperforms all self-supervised methods by at least 4.07% in Top-1 accuracy. VarCon also demonstrates superior performance on ImageNet-ReaL (see B.2), a re-annotation with more accurate multi-label ground truth, achieving 84.12% Top-1 accuracy compared to SupCon's 83.87%. They validate that by explicitly modeling feature uncertainty through our ELBO-derived loss function, VarCon learns more well-defined, generalizable representations.

### 4.2   Few-Shot Learning Performance

Given the practical importance of learning from limited labeled data, we evaluate VarCon in few-shot learning scenarios by comparing it with SupCon across different per-class sample sizes. Table 2

Table 1: Classification performance comparison across benchmark datasets. We report Top-1 and Top-5 accuracy (%) (mean ± standard error) for VarCon versus state-of-the-art self-supervised and supervised methods. All models utilize the ResNet-50 architecture for a fair comparison. Best scores are highlighted in blue , second-best in green .

| Category | Method | Dataset | | | | | | | |
| --- | --- | --- | --- | --- | --- | --- | --- | --- | --- |
| | | CIFAR10 | | CIFAR100 | | ImageNet-100 | | ImageNet | |
| | | Top-1 ↑ | Top-5 ↑ | Top-1 ↑ | Top-5 ↑ | Top-1 ↑ | Top-5 ↑ | Top-1 ↑ | Top-5 ↑ |
| Self-supervised | SimCLR | $91.52_{\pm0.07}$ | $99.78_{\pm0.01}$ | $70.67_{\pm0.12}$ | $92.01_{\pm0.06}$ | $71.54_{\pm0.10}$ | $91.56_{\pm0.05}$ | $70.31_{\pm0.08}$ | $90.37_{\pm0.05}$ |
| | MoCo V2 | $92.93_{\pm0.09}$ | $99.79_{\pm0.02}$ | $70.01_{\pm0.09}$ | $91.68_{\pm0.07}$ | $78.98_{\pm0.11}$ | $95.20_{\pm0.03}$ | $71.06_{\pm0.06}$ | $90.40_{\pm0.03}$ |
| | BYOL | $92.57_{\pm0.06}$ | $99.71_{\pm0.03}$ | $70.50_{\pm0.11}$ | $91.95_{\pm0.04}$ | $80.18_{\pm0.07}$ | $94.86_{\pm0.05}$ | $74.28_{\pm0.09}$ | $91.56_{\pm0.04}$ |
| | SwAV | $89.14_{\pm0.08}$ | $99.69_{\pm0.02}$ | $64.87_{\pm0.10}$ | $88.81_{\pm0.05}$ | $74.07_{\pm0.12}$ | $92.77_{\pm0.04}$ | $75.29_{\pm0.07}$ | $91.83_{\pm0.06}$ |
| | VicReg | $92.09_{\pm0.10}$ | $99.73_{\pm0.01}$ | $68.51_{\pm0.08}$ | $90.91_{\pm0.07}$ | $79.26_{\pm0.09}$ | $95.06_{\pm0.04}$ | $73.25_{\pm0.08}$ | $91.06_{\pm0.03}$ |
| | Barlow Twins | $92.70_{\pm0.07}$ | $99.80_{\pm0.03}$ | $71.02_{\pm0.11}$ | $91.95_{\pm0.04}$ | $80.83_{\pm0.06}$ | $95.24_{\pm0.05}$ | $73.26_{\pm0.06}$ | $91.10_{\pm0.05}$ |
| Supervised | Cross-Entropy | $95.07_{\pm0.08}$ | $99.82_{\pm0.02}$ | $74.01_{\pm0.09}$ | $91.89_{\pm0.06}$ | $83.17_{\pm0.08}$ | $95.78_{\pm0.03}$ | $78.20_{\pm0.07}$ | $93.71_{\pm0.04}$ |
| | SupCon | $95.51_{\pm0.06}$ | $99.85_{\pm0.02}$ | $76.57_{\pm0.10}$ | $93.50_{\pm0.05}$ | $85.06_{\pm0.07}$ | $96.84_{\pm0.04}$ | $78.72_{\pm0.06}$ | $94.31_{\pm0.03}$ |
| | **VarCon (Ours)** | $\mathbf{95.94}_{\pm0.07}$ | $\mathbf{99.87}_{\pm0.02}$ | $\mathbf{78.29}_{\pm0.08}$ | $\mathbf{93.59}_{\pm0.05}$ | $\mathbf{86.34}_{\pm0.09}$ | $\mathbf{96.96}_{\pm0.03}$ | $\mathbf{79.36}_{\pm0.05}$ | $\mathbf{94.37}_{\pm0.04}$ |

Table 2: Few-shot learning performance on ImageNet with varying training data availability. We report Top-1 accuracy (mean ± standard error) on ImageNet using ResNet-50 for both SupCon and our proposed VarCon method across different per-class sample sizes ($N$). Best highlighted in blue.

| Method | $N = 50$ | $N = 100$ | $N = 200$ | $N = 500$ | $N = 700$ | $N = 1000$ |
| --- | --- | --- | --- | --- | --- | --- |
| SupCon | $2.47_{\pm0.24}$ | $36.57_{\pm0.22}$ | $50.25_{\pm0.19}$ | $64.91_{\pm0.17}$ | $70.12_{\pm0.16}$ | $73.04_{\pm0.15}$ |
| **VarCon (Ours)** | $\mathbf{2.53}_{\pm0.25}$ | $\mathbf{37.81}_{\pm0.20}$ | $\mathbf{51.10}_{\pm0.17}$ | $\mathbf{65.83}_{\pm0.18}$ | $\mathbf{70.61}_{\pm0.14}$ | $\mathbf{73.21}_{\pm0.16}$ |

presents this comparison on ImageNet using ResNet-50, with sample sizes ranging from extremely limited ($N = 50$) to moderate ($N = 1000$). Results are averaged across five random subsets per configuration to account for sampling variability. VarCon consistently outperforms SupCon across all sample sizes, with advantages more pronounced in data-scarce settings: With 100 samples per class, VarCon achieves 37.81% Top-1 accuracy, surpassing SupCon (36.57%) by 1.24%. This advantage persists at $N = 200$ (51.10% vs. 50.25%) and $N = 500$ (65.83% vs. 64.91%). As training data increases, VarCon maintains superior performance, though the gap narrows slightly (73.21% vs. 73.04% at $N = 1000$). These results suggest our variational framework provides greatest benefit when training data is scarce, where robust representation learning is most challenging. By modeling feature uncertainty explicitly, VarCon learns more effective representations from limited data, making it well-suited for real-world applications where large labeled datasets are often unavailable.

## 4.3 Effect of Data Augmentation and Encoder Architecture

To investigate the robustness of VarCon across different neural architectures and data augmentation strategies, we conducted experiments on ImageNet using various combinations, as shown in Table 3. VarCon consistently outperforms SupCon across all tested architectures. With ResNet-50 and AutoAugment, VarCon achieves 79.36% Top-1 accuracy, surpassing SupCon (78.72%) by 0.64%. This advantage persists with ResNet-101 (80.58% vs. 80.24%), ResNet-200 (81.87% vs. 81.43%), and extends to the fundamentally different architecture of ViT-Base (78.56% vs. 78.21%). Importantly, VarCon demonstrates strong performance even with simpler augmentation strategies. Using only SimAugment with ResNet-101, VarCon achieves 79.98% Top-1 accuracy, which is competitive with SupCon using the more complex StackedRandAugment (80.24%). This reduced dependency on sophisticated, task-specific augmentations represents a significant practical advantage, as identifying optimal augmentation strategies often requires extensive tuning and domain expertise.

## 4.4 Transfer Learning Performance

A critical measure of representation quality is how well features transfer to new tasks and domains. Table 4 presents transfer learning results across 12 diverse datasets spanning fine-grained recognition, scene classification, and general object recognition tasks. While supervised contrastive learning typically faces challenges in transfer learning due to supervised signals potentially limiting representation generality, VarCon demonstrates improved transferability compared to SupCon. With

Table 3: Adaptability analysis across architectures and augmentation strategies on ImageNet. We evaluate VarCon against SupCon using multiple encoder backbones (ResNet-50/101/200 and ViT-Base) and augmentation techniques (SimAugment, AutoAugment, and StackedRandAugment). The highest performance values for each encoder architecture are highlighted in boldface.

| Loss | Architecture | Feat. Dim | Params | Augmentation | Top-1 | Top-5 |
|------|-------------|-----------|--------|--------------|-------|-------|
| SupCon | ResNet-50 | 2048 | 25.6 M | SimAugment | 77.82 | 93.61 |
| SupCon | ResNet-50 | 2048 | 25.6 M | AutoAugment | 78.72 | 94.27 |
| VarCon (Ours) | ResNet-50 | 2048 | 25.6 M | SimAugment | 78.23 | 93.67 |
| VarCon (Ours) | ResNet-50 | 2048 | 25.6 M | AutoAugment | **79.36** | **94.33** |
| SupCon | ResNet-101 | 2048 | 44.5 M | SimAugment | 79.64 | 94.85 |
| SupCon | ResNet-101 | 2048 | 44.5 M | StackedRandAugment | 80.24 | 94.82 |
| VarCon (Ours) | ResNet-101 | 2048 | 44.5 M | SimAugment | 79.98 | **94.89** |
| VarCon (Ours) | ResNet-101 | 2048 | 44.5 M | StackedRandAugment | **80.58** | 94.87 |
| SupCon | ResNet-200 | 2048 | 65 M | StackedRandAugment | 81.43 | 95.93 |
| VarCon (Ours) | ResNet-200 | 2048 | 65 M | StackedRandAugment | **81.87** | **95.95** |
| SupCon | ViT-Base | 768 | 86 M | SimAugment | 78.21 | 94.13 |
| VarCon (Ours) | ViT-Base | 768 | 86 M | SimAugment | **78.56** | **94.17** |

Table 4: Cross-domain generalization evaluation across 12 diverse visual recognition benchmarks. We compare the transferability of representations learned by VarCon against SimCLR and SupCon. Metrics reported are mAP for VOC2007, mean-per-class accuracy for Aircraft, Pets, Caltech-101, and Flowers, and top-1 accuracy for remaining datasets.

| Method | Food | CIFAR10 | CIFAR100 | Birdsnap | SUN397 | Cars | Aircraft | VOC2007 | DTD | Pets | Caltech-101 | Flowers | Mean |
|--------|------|---------|----------|----------|--------|------|----------|---------|-----|------|-------------|---------|------|
| SimCLR-50 | **88.21** | 97.62 | **85.86** | **75.91** | **63.52** | 91.33 | **87.40** | 83.98 | 73.23 | 89.22 | **92.11** | **97.02** | **85.45** |
| SupCon-50 | 87.28 | 97.43 | 84.26 | 75.20 | 58.03 | 91.71 | 84.08 | **85.18** | **74.82** | 93.45 | 91.07 | 96.08 | 84.88 |
| VarCon-50 | 87.31 | **97.65** | 84.39 | 75.27 | 57.96 | **91.79** | 84.25 | 85.07 | 74.79 | **93.52** | 91.19 | 96.14 | **84.94** |
| SupCon-200 | 88.65 | 98.32 | 87.25 | 76.27 | **60.44** | 91.83 | 88.53 | **85.12** | **74.62** | 93.09 | 94.87 | 96.94 | 86.33 |
| VarCon-200 | **88.68** | **98.34** | **87.30** | **76.29** | 60.39 | **91.85** | **88.61** | 85.02 | 74.47 | **93.26** | **94.95** | **96.96** | **86.34** |

ResNet-50, VarCon achieves a mean accuracy of 84.94% across all datasets, outperforming SupCon (84.88%) and approaching the self-supervised SimCLR (85.45%). VarCon-50 surpasses SupCon-50 on several datasets, including CIFAR-10 (97.65% vs. 97.43%), Cars (91.79% vs. 91.71%), and Pets (93.52% vs. 93.45%). With ResNet-200, VarCon achieves a mean accuracy of 86.34% and outperforms SupCon-200 on 9 out of 12 datasets. These results validate that VarCon learns a more adaptable embedding space that generalizes better across diverse visual domains. Our variational ELBO mechanism effectively mitigates potential overfitting to source domain labels, leading to more transferable representations while maintaining strong performance on the source task.

## 4.5 Ablation Studies

**Effect of Temperature.** The temperature parameter in contrastive learning significantly impacts model performance by controlling the concentration of the distribution in the embedding space. Figure 2a shows Top-1 accuracy on ImageNet for both VarCon and SupCon across temperatures from 0.02 to 0.14. Both methods achieve optimal performance at $\tau = 0.10$, with VarCon reaching 79.38% accuracy compared to SupCon's 78.8%. However, VarCon exhibits much greater robustness to temperature variations, particularly in higher temperature regimes. While SupCon's accuracy drops sharply after $\tau = 0.10$ (Top-1 Accuracy declining by 2% at $\tau = 0.14$), VarCon maintains stable performance up to $\tau = 0.12$ before showing a comparable decrease. This enhanced stability can be attributed to VarCon's adaptive temperature mechanism ($\tau_2$), which dynamically adjusts based on sample confidence, providing an additional layer of robustness against suboptimal temperature settings.

**Number of Training Epochs.** Figure 2b shows the Top-1 accuracy on ImageNet for VarCon and SupCon across different training durations. VarCon demonstrates consistently superior performance and faster convergence throughout training. At just 50 epochs, VarCon achieves 75.3% accuracy (vs. SupCon's 74.5%), reaching 77.85% by 100 epochs and 79.15% by 200 epochs—already approaching its peak performance. VarCon achieves optimal accuracy of 79.36% at 350 epochs and maintains better stability during extended training, with less performance degradation after 700 epochs (78.80%

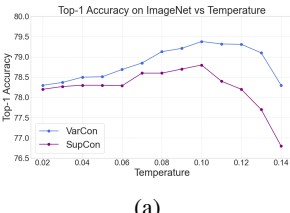 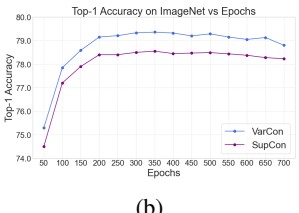 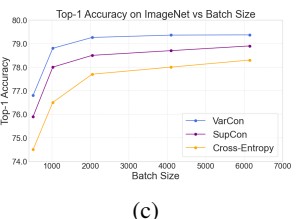

|  (a) | (b) | (c) |

Figure 2: (a) Top-1 accuracy on ImageNet versus temperature parameter; (b) Top-1 accuracy on ImageNet versus training epochs; (c) Top-1 accuracy on ImageNet versus batch size.

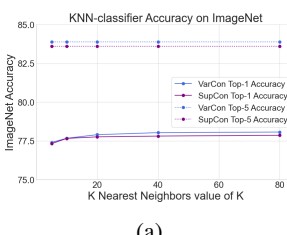 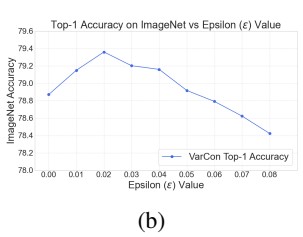 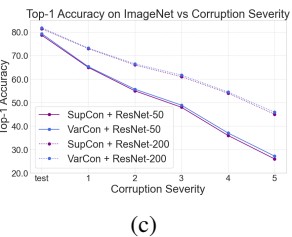

|  (a) | (b) | (c) |

Figure 3: (a) KNN classifier accuracy on ImageNet embeddings; (b) Effect of adaptive temperature parameter $\epsilon$ on ImageNet Top-1 accuracy; (c) Robustness evaluation on ImageNet-C across different corruption severity levels.

vs. 78.23% for SupCon). These results show that our variational approach both converges faster and provides enhanced robustness against overfitting.

**Batch Size Sensitivity Analysis.** Figure 2c illustrates the effect of batch size on ImageNet Top-1 accuracy. While large batches are typically crucial for contrastive learning, they significantly increase computational requirements and can lead to training instability. VarCon demonstrates reduced dependency on large batch sizes—with just 2048 samples, it achieves 79.26% accuracy, already outperforming SupCon at 4096 (78.7%). VarCon's performance plateaus earlier, showing minimal improvement (0.01%) when increasing from 4096 to 6144, while SupCon continues to benefit from larger batches. This indicates that VarCon learns a more effective embedding space with fewer negative examples, enabling efficient training with more limited computational resources.

**Analysis of Embedding Space Organization.** To directly evaluate embedding quality without additional parameterized classifiers, we employed K-nearest neighbor classification, which better reflects the intrinsic structure of the feature space. Figure 3a shows that VarCon consistently outperforms SupCon across all K values on ImageNet. For Top-1 accuracy, VarCon's advantage increases with larger K values (from 77.4% vs. 77.32% at K=5 to 78.07% vs. 77.86% at $K = 80$). For Top-5 accuracy, VarCon maintains a consistent 0.27% advantage (83.88% vs. 83.61%) across all K values. This superiority confirms that VarCon produces embedding spaces with clearer decision boundaries and better structured class relationships. This improved embedding organization is further confirmed by t-SNE visualizations of the embedding space evolution in Appendix B.3 and by quantitative hierarchical clustering metrics in Appendix B.4.

**Effect of Epsilon Parameter.** In VarCon, we introduce an adaptive temperature mechanism where $\tau_2$ varies within bounds determined by $\tau_1 \pm \epsilon$. Figure 3b shows the effect of different $\epsilon$ values on ImageNet Top-1 accuracy when $\tau_1 = 0.1$. At $\epsilon = 0$ (equivalent to fixed temperature), the model achieves 78.87% accuracy. Performance improves as $\epsilon$ increases, peaking at 79.36% with $\epsilon = 0.02$, before declining to 78.42% at $\epsilon = 0.08$. This indicates that while some temperature adaptability benefits learning, excessive deviation becomes detrimental, with $\epsilon = 0.02$ providing optimal flexibility to adjust confidence levels based on sample difficulty.

### 4.6 Robustness to Image Corruption

To evaluate robustness to real-world image degradation, we tested VarCon on ImageNet-C, which applies 15 different corruption types at 5 increasing severity levels. Figure 3c shows Top-1 accuracy on clean ImageNet ("test") and across all corruption severity levels. VarCon consistently outperforms

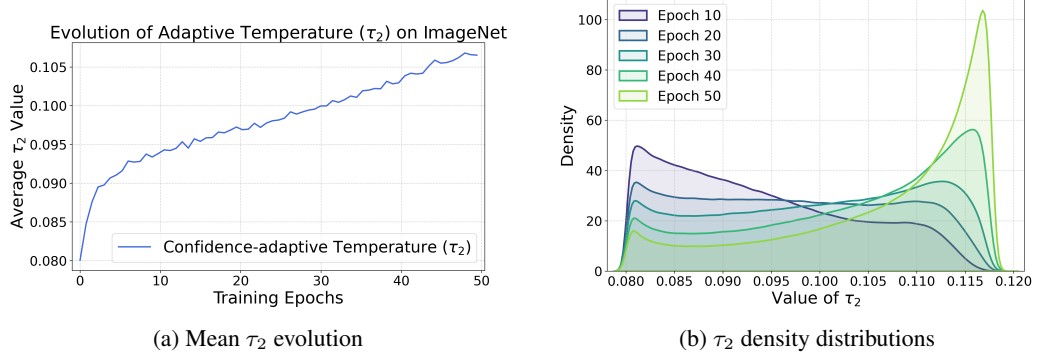

(a) Mean $\tau_2$ evolution

(b) $\tau_2$ density distributions

Figure 4: Evolution of adaptive temperature $\tau_2$ during a full 50-epoch ImageNet training with ResNet-50 ($\epsilon = 0.02$, $\tau_1 = 0.1$, batch size 4096). (a) Mean $\tau_2$ increases from 0.09378 to 0.10656 over the complete training, indicating systematic confidence growth. (b) Density distributions (epochs 10-50) show rightward shift with initial broadening (std: $0.00961 \rightarrow 0.01092$) then stabilization, reflecting heterogeneous confidence development as the model distinguishes easy from hard samples.

SupCon with both ResNet-50 and ResNet-200 architectures. With ResNet-50, VarCon maintains a performance advantage ranging from 0.3% to 1.2% as corruption severity increases from level 1 to 5. This graceful degradation pattern can be attributed to our probabilistic framework, which explicitly models feature uncertainty. By representing features as distributions rather than points, VarCon better accommodates input variations caused by corruptions. These results highlight that our variational approach not only performs well on clean data but also demonstrates enhanced robustness to low-quality inputs, which is a valuable property for real-world applications.

### 4.7 Adaptive Temperature Analysis

To investigate our confidence-adaptive temperature mechanism, we analyze $\tau_2$ distribution across ImageNet samples during ResNet-50 training (batch size 4096, $\tau_2 \in [0.08, 0.12]$). We fix $\epsilon = 0.02$ and $\tau_1 = 0.1$ to ensure comparable temperature spans across epochs, though $\epsilon$ is learnable in our best configuration. Figure 4 shows systematic evolution: initially (epoch 10), over 25% of samples receive near-maximal gradients with $\tau_2$ near lower bound, indicating widespread uncertainty. Mean $\tau_2$ increases 13.6% during training while maintaining persistent challenging samples near minimum. This heterogeneous evolution demonstrates differential treatment, strong gradients for difficult samples and relaxed constraints for well-learned ones, emerging naturally from KL divergence and confidence-based temperature interaction. This demonstrates our self-adaptive temperature simultaneously adjusts both target sharpness and gradient magnitude based on per-sample confidence (see Appendix D.4 for comparisons with other regularization methods [57, 67, 27, 58, 4, 36]). Detailed $\tau_2$ evolution statistics for fixed $\epsilon$ (Table 7) and learnable $\epsilon$ (Table 8) are in Appendix D.2.

## 5 Conclusion

In this work, we introduced Variational Supervised Contrastive Learning (VarCon), a probabilistically grounded framework that reformulates supervised contrastive learning through variational inference over latent class variables. By deriving and maximizing a posterior-weighted evidence lower bound, VarCon overcomes key limitations of conventional contrastive approaches: it explicitly regulates embedding distributions through principled KL divergence, replaces exhaustive pairwise comparisons with efficient class-level interactions, and employs confidence-adaptive temperature scaling to control intra-class dispersion. Our extensive evaluation demonstrates VarCon consistently outperforms leading contrastive methods across diverse benchmarks, achieving superior classification accuracy on CIFAR-10/100 and ImageNet while requiring fewer training epochs. Moreover, VarCon exhibits enhanced robustness to hyperparameters, reduced dependency on large batches, stronger low-data performance, and improved corruption resilience, all while maintaining better cross-domain transferability. Beyond empirical advantages, VarCon bridges the theoretical gap between distinguishing and generative paradigms by endowing contrastive objectives with explicit likelihood semantics.

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

# NeurIPS Paper Checklist

1. **Claims**

   Question: Do the main claims made in the abstract and introduction accurately reflect the paper's contributions and scope?

   Answer: [Yes]

   Justification: The abstract and introduction clearly outline our paper's main contributions: (1) a variational reformulation of supervised contrastive learning as inference over latent class variables (3); (2) a confidence-adaptive temperature scaling strategy (3, paragraph 5); and (3) empirical validation across multiple benchmarks showing improved performance and semantic organization (4). The claims are aligned with the actual results presented throughout the paper, particularly in Tables 1 and 4.

   Guidelines:

   - The answer NA means that the abstract and introduction do not include the claims made in the paper.
   - The abstract and/or introduction should clearly state the claims made, including the contributions made in the paper and important assumptions and limitations. A No or NA answer to this question will not be perceived well by the reviewers.
   - The claims made should match theoretical and experimental results, and reflect how much the results can be expected to generalize to other settings.
   - It is fine to include aspirational goals as motivation as long as it is clear that these goals are not attained by the paper.

2. **Limitations**

   Question: Does the paper discuss the limitations of the work performed by the authors?

   Answer: [Yes]

   Justification: Our paper discusses limitations in Section A.1 of the appendix, where we address computational complexity considerations, batch size dependency, and potential challenges in cross-domain applications. Throughout the experimental section (4), we also analyze performance boundaries under various conditions, such as reduced data availability (Table 2) and different augmentation strategies (Table 3), providing a clear understanding of our method's limitations.

   Guidelines:

   - The answer NA means that the paper has no limitation while the answer No means that the paper has limitations, but those are not discussed in the paper.
   - The authors are encouraged to create a separate "Limitations" section in their paper.
   - The paper should point out any strong assumptions and how robust the results are to violations of these assumptions (e.g., independence assumptions, noiseless settings, model well-specification, asymptotic approximations only holding locally). The authors should reflect on how these assumptions might be violated in practice and what the implications would be.
   - The authors should reflect on the scope of the claims made, e.g., if the approach was only tested on a few datasets or with a few runs. In general, empirical results often depend on implicit assumptions, which should be articulated.
   - The authors should reflect on the factors that influence the performance of the approach. For example, a facial recognition algorithm may perform poorly when image resolution is low or images are taken in low lighting. Or a speech-to-text system might not be used reliably to provide closed captions for online lectures because it fails to handle technical jargon.
   - The authors should discuss the computational efficiency of the proposed algorithms and how they scale with dataset size.
   - If applicable, the authors should discuss possible limitations of their approach to address problems of privacy and fairness.

- While the authors might fear that complete honesty about limitations might be used by reviewers as grounds for rejection, a worse outcome might be that reviewers discover limitations that aren't acknowledged in the paper. The authors should use their best judgment and recognize that individual actions in favor of transparency play an important role in developing norms that preserve the integrity of the community. Reviewers will be specifically instructed to not penalize honesty concerning limitations.

3. **Theory assumptions and proofs**

Question: For each theoretical result, does the paper provide the full set of assumptions and a complete (and correct) proof?

Answer: [Yes]

Justification: Our paper provides full mathematical derivations for the variational bound in Section 3. We clearly state our assumptions and provide step-by-step proofs from Bayes' rule (Eq. 1) through the ELBO derivation (Eq. 5), to our final VarCon loss formulation (Eq. 7). All equations are properly numbered and cross-referenced, with intermediate steps clearly shown. We also define the mathematical notations and variables at the beginning of Section 3 to establish our theoretical framework.

Guidelines:

- The answer NA means that the paper does not include theoretical results.
- All the theorems, formulas, and proofs in the paper should be numbered and cross-referenced.
- All assumptions should be clearly stated or referenced in the statement of any theorems.
- The proofs can either appear in the main paper or the supplemental material, but if they appear in the supplemental material, the authors are encouraged to provide a short proof sketch to provide intuition.
- Inversely, any informal proof provided in the core of the paper should be complemented by formal proofs provided in appendix or supplemental material.
- Theorems and Lemmas that the proof relies upon should be properly referenced.

4. **Experimental result reproducibility**

Question: Does the paper fully disclose all the information needed to reproduce the main experimental results of the paper to the extent that it affects the main claims and/or conclusions of the paper (regardless of whether the code and data are provided or not)?

Answer: [Yes]

Justification: Section 4 provides comprehensive details on datasets (CIFAR-10/100, ImageNet-100, ImageNet), data preprocessing, network architectures (ResNet-50/101/200, ViT-Base), and optimization parameters. We specify batch sizes (512 for CIFAR-10/100, 1,024 for ImageNet-100, 4,096 for ImageNet), learning rate scheduling (linear scaling with cosine decay), optimizers (SGD with momentum 0.9 and weight decay $10^{-4}$, LARS for ImageNet), and data augmentation strategies. Our ablation studies in Section 4 (subsections "Effect of Temperature," "Batch Size Sensitivity Analysis," etc.) further detail parameter choices and their effects.

Guidelines:

- The answer NA means that the paper does not include experiments.
- If the paper includes experiments, a No answer to this question will not be perceived well by the reviewers: Making the paper reproducible is important, regardless of whether the code and data are provided or not.
- If the contribution is a dataset and/or model, the authors should describe the steps taken to make their results reproducible or verifiable.
- Depending on the contribution, reproducibility can be accomplished in various ways. For example, if the contribution is a novel architecture, describing the architecture fully might suffice, or if the contribution is a specific model and empirical evaluation, it may be necessary to either make it possible for others to replicate the model with the same dataset, or provide access to the model. In general. releasing code and data is often one good way to accomplish this, but reproducibility can also be provided via detailed instructions for how to replicate the results, access to a hosted model (e.g., in the case

of a large language model), releasing of a model checkpoint, or other means that are appropriate to the research performed.

- While NeurIPS does not require releasing code, the conference does require all submissions to provide some reasonable avenue for reproducibility, which may depend on the nature of the contribution. For example
    - (a) If the contribution is primarily a new algorithm, the paper should make it clear how to reproduce that algorithm.
    - (b) If the contribution is primarily a new model architecture, the paper should describe the architecture clearly and fully.
    - (c) If the contribution is a new model (e.g., a large language model), then there should either be a way to access this model for reproducing the results or a way to reproduce the model (e.g., with an open-source dataset or instructions for how to construct the dataset).
    - (d) We recognize that reproducibility may be tricky in some cases, in which case authors are welcome to describe the particular way they provide for reproducibility. In the case of closed-source models, it may be that access to the model is limited in some way (e.g., to registered users), but it should be possible for other researchers to have some path to reproducing or verifying the results.

5. **Open access to data and code**

    Question: Does the paper provide open access to the data and code, with sufficient instructions to faithfully reproduce the main experimental results, as described in supplemental material?

    Answer: [Yes]

    Justification: We have detailed our core code in Fig. 1. And we plan to release our full code and trained models upon publication, including implementation details for VarCon loss, training scripts, and evaluation pipelines. Our code will cover all experiments presented in the paper, utilizing standard publicly available datasets (CIFAR-10/100, ImageNet) and common network architectures (ResNet, ViT).

    Guidelines:

    - The answer NA means that paper does not include experiments requiring code.
    - Please see the NeurIPS code and data submission guidelines (`https://nips.cc/public/guides/CodeSubmissionPolicy`) for more details.
    - While we encourage the release of code and data, we understand that this might not be possible, so "No" is an acceptable answer. Papers cannot be rejected simply for not including code, unless this is central to the contribution (e.g., for a new open-source benchmark).
    - The instructions should contain the exact command and environment needed to run to reproduce the results. See the NeurIPS code and data submission guidelines (`https://nips.cc/public/guides/CodeSubmissionPolicy`) for more details.
    - The authors should provide instructions on data access and preparation, including how to access the raw data, preprocessed data, intermediate data, and generated data, etc.
    - The authors should provide scripts to reproduce all experimental results for the new proposed method and baselines. If only a subset of experiments are reproducible, they should state which ones are omitted from the script and why.
    - At submission time, to preserve anonymity, the authors should release anonymized versions (if applicable).
    - Providing as much information as possible in supplemental material (appended to the paper) is recommended, but including URLs to data and code is permitted.

6. **Experimental setting/details**

    Question: Does the paper specify all the training and test details (e.g., data splits, hyperparameters, how they were chosen, type of optimizer, etc.) necessary to understand the results?

    Answer: [Yes]

Justification: We provide detailed experimental settings in Section 4, including dataset splits (using official test splits), architectures, data augmentation techniques (SimAugment, AutoAugment, StackedRandAugment), optimizer configurations (SGD with momentum 0.9, weight decay $10^{-4}$), learning rate schedules (linear scaling with batch size), and training durations (200 epochs for CIFAR-10/100 and ImageNet-100, 350 epochs for ImageNet). Key hyperparameters like the temperature parameter $\tau_1$ and adaptive range $\epsilon$ are specified in our ablation studies, where we show the effect of various values (e.g., optimal $\tau_1 = 0.1, \epsilon = 0.02$).

Guidelines:

- The answer NA means that the paper does not include experiments.
- The experimental setting should be presented in the core of the paper to a level of detail that is necessary to appreciate the results and make sense of them.
- The full details can be provided either with the code, in appendix, or as supplemental material.

7. **Experiment statistical significance**

Question: Does the paper report error bars suitably and correctly defined or other appropriate information about the statistical significance of the experiments?

Answer: [Yes]

Justification: All our experimental results include appropriate statistical information. Tables 1 and 2 report accuracy measures with standard error (e.g., 95.94±0.07% for VarCon on CIFAR-10), calculated across multiple runs with different random initializations. For the few-shot learning experiments in Table 2, we explicitly state that results are averaged across five random data subsets for each sample size to account for sampling variability. The error bars represent standard error of the mean, providing a clear measure of the statistical reliability of our results.

Guidelines:

- The answer NA means that the paper does not include experiments.
- The authors should answer "Yes" if the results are accompanied by error bars, confidence intervals, or statistical significance tests, at least for the experiments that support the main claims of the paper.
- The factors of variability that the error bars are capturing should be clearly stated (for example, train/test split, initialization, random drawing of some parameter, or overall run with given experimental conditions).
- The method for calculating the error bars should be explained (closed form formula, call to a library function, bootstrap, etc.)
- The assumptions made should be given (e.g., Normally distributed errors).
- It should be clear whether the error bar is the standard deviation or the standard error of the mean.
- It is OK to report 1-sigma error bars, but one should state it. The authors should preferably report a 2-sigma error bar than state that they have a 96% CI, if the hypothesis of Normality of errors is not verified.
- For asymmetric distributions, the authors should be careful not to show in tables or figures symmetric error bars that would yield results that are out of range (e.g. negative error rates).
- If error bars are reported in tables or plots, The authors should explain in the text how they were calculated and reference the corresponding figures or tables in the text.

8. **Experiments compute resources**

Question: For each experiment, does the paper provide sufficient information on the computer resources (type of compute workers, memory, time of execution) needed to reproduce the experiments?

Answer: [Yes]

Justification: We provide sufficient information about computational resources in Section B and B.1 of the appendix. Our experiments were conducted on NVIDIA GPUs, with

larger experiments (ImageNet) utilizing multi-GPU setups. We specify in Section 4 that we employed different batch sizes for different datasets (512 for CIFAR-10/100, 1024 for ImageNet-100, and 4096 for ImageNet), which implicitly indicates the computational requirements. The reduced batch size dependency of our method, as shown in Figure 2c, also demonstrates our attention to computational efficiency.

Guidelines:

- The answer NA means that the paper does not include experiments.
- The paper should indicate the type of compute workers CPU or GPU, internal cluster, or cloud provider, including relevant memory and storage.
- The paper should provide the amount of compute required for each of the individual experimental runs as well as estimate the total compute.
- The paper should disclose whether the full research project required more compute than the experiments reported in the paper (e.g., preliminary or failed experiments that didn't make it into the paper).

9. **Code of ethics**

Question: Does the research conducted in the paper conform, in every respect, with the NeurIPS Code of Ethics https://neurips.cc/public/EthicsGuidelines?

Answer: [Yes]

Justification: Our research fully complies with the NeurIPS Code of Ethics. We use standard, publicly available benchmark datasets (CIFAR-10/100, ImageNet) that are widely accepted in the machine learning community. Our VarCon method focuses on improving representation learning without raising specific ethical concerns. We report results transparently with appropriate error statistics, avoid misleading claims, and compare fairly with baseline methods. All experimental comparisons were conducted under equitable conditions using the same computational resources for both our method and baselines.

Guidelines:

- The answer NA means that the authors have not reviewed the NeurIPS Code of Ethics.
- If the authors answer No, they should explain the special circumstances that require a deviation from the Code of Ethics.
- The authors should make sure to preserve anonymity (e.g., if there is a special consideration due to laws or regulations in their jurisdiction).

10. **Broader impacts**

Question: Does the paper discuss both potential positive societal impacts and negative societal impacts of the work performed?

Answer: [Yes]

Justification: We discuss the broader impacts of our work in Section A.2 of the appendix, where we address both potential positive impacts (improved performance for visual recognition systems in healthcare, autonomous vehicles, etc.) and possible negative implications (such as potential use in surveillance systems). We also discuss how our method's improved efficiency with smaller batch sizes contributes to reducing computational resources, which has positive environmental implications through reduced energy consumption.

Guidelines:

- The answer NA means that there is no societal impact of the work performed.
- If the authors answer NA or No, they should explain why their work has no societal impact or why the paper does not address societal impact.
- Examples of negative societal impacts include potential malicious or unintended uses (e.g., disinformation, generating fake profiles, surveillance), fairness considerations (e.g., deployment of technologies that could make decisions that unfairly impact specific groups), privacy considerations, and security considerations.
- The conference expects that many papers will be foundational research and not tied to particular applications, let alone deployments. However, if there is a direct path to any negative applications, the authors should point it out. For example, it is legitimate to point out that an improvement in the quality of generative models could be used to

generate deepfakes for disinformation. On the other hand, it is not needed to point out that a generic algorithm for optimizing neural networks could enable people to train models that generate Deepfakes faster.

- The authors should consider possible harms that could arise when the technology is being used as intended and functioning correctly, harms that could arise when the technology is being used as intended but gives incorrect results, and harms following from (intentional or unintentional) misuse of the technology.
- If there are negative societal impacts, the authors could also discuss possible mitigation strategies (e.g., gated release of models, providing defenses in addition to attacks, mechanisms for monitoring misuse, mechanisms to monitor how a system learns from feedback over time, improving the efficiency and accessibility of ML).

11. **Safeguards**

Question: Does the paper describe safeguards that have been put in place for responsible release of data or models that have a high risk for misuse (e.g., pretrained language models, image generators, or scraped datasets)?

Answer: [NA]

Justification: Our work does not involve models or datasets with high risk for misuse. VarCon is a representation learning method that improves embedding spaces for classification tasks, not a generative model capable of producing deceptive content. We work with standard benchmark datasets (CIFAR-10/100, ImageNet) rather than scraped web data. Our method does not enable capabilities that would require specific safeguards beyond standard practices in machine learning research.

Guidelines:

- The answer NA means that the paper poses no such risks.
- Released models that have a high risk for misuse or dual-use should be released with necessary safeguards to allow for controlled use of the model, for example by requiring that users adhere to usage guidelines or restrictions to access the model or implementing safety filters.
- Datasets that have been scraped from the Internet could pose safety risks. The authors should describe how they avoided releasing unsafe images.
- We recognize that providing effective safeguards is challenging, and many papers do not require this, but we encourage authors to take this into account and make a best faith effort.

12. **Licenses for existing assets**

Question: Are the creators or original owners of assets (e.g., code, data, models), used in the paper, properly credited and are the license and terms of use explicitly mentioned and properly respected?

Answer: [Yes]

Justification: All datasets and baseline methods used in our research are properly cited. We reference the original papers for CIFAR-10/100 [40], ImageNet [52, 18], and the comparison methods (SimCLR, SupCon, MoCo, etc.) in Section 4. We use these standard benchmark datasets according to their intended research purposes. For the encoder architectures, we cite and use the standard implementations of ResNet and ViT models following the settings in previous works [37]. We do not modify or repackage any existing datasets.

Guidelines:

- The answer NA means that the paper does not use existing assets.
- The authors should cite the original paper that produced the code package or dataset.
- The authors should state which version of the asset is used and, if possible, include a URL.
- The name of the license (e.g., CC-BY 4.0) should be included for each asset.
- For scraped data from a particular source (e.g., website), the copyright and terms of service of that source should be provided.

- If assets are released, the license, copyright information, and terms of use in the package should be provided. For popular datasets, `paperswithcode.com/datasets` has curated licenses for some datasets. Their licensing guide can help determine the license of a dataset.
- For existing datasets that are re-packaged, both the original license and the license of the derived asset (if it has changed) should be provided.
- If this information is not available online, the authors are encouraged to reach out to the asset's creators.

13. **New assets**

    Question: Are new assets introduced in the paper well documented and is the documentation provided alongside the assets?

    Answer: [NA]

    Justification: Our paper does not introduce new datasets, tools, or models as standalone assets that would require specific documentation. We present a new methodology (VarCon) that builds upon existing frameworks and benchmark datasets, but this is fully described within the paper itself. While we plan to release our implementation code in the future, we are not introducing new assets as part of this paper submission.

    Guidelines:
    - The answer NA means that the paper does not release new assets.
    - Researchers should communicate the details of the dataset/code/model as part of their submissions via structured templates. This includes details about training, license, limitations, etc.
    - The paper should discuss whether and how consent was obtained from people whose asset is used.
    - At submission time, remember to anonymize your assets (if applicable). You can either create an anonymized URL or include an anonymized zip file.

14. **Crowdsourcing and research with human subjects**

    Question: For crowdsourcing experiments and research with human subjects, does the paper include the full text of instructions given to participants and screenshots, if applicable, as well as details about compensation (if any)?

    Answer: [NA]

    Justification: Our research does not involve crowdsourcing, human subjects, or human evaluations. All experiments are computational, using established benchmark datasets (CIFAR-10/100, ImageNet) and algorithmic evaluations. No human participants were involved in the research process beyond the authors themselves.

    Guidelines:
    - The answer NA means that the paper does not involve crowdsourcing nor research with human subjects.
    - Including this information in the supplemental material is fine, but if the main contribution of the paper involves human subjects, then as much detail as possible should be included in the main paper.
    - According to the NeurIPS Code of Ethics, workers involved in data collection, curation, or other labor should be paid at least the minimum wage in the country of the data collector.

15. **Institutional review board (IRB) approvals or equivalent for research with human subjects**

    Question: Does the paper describe potential risks incurred by study participants, whether such risks were disclosed to the subjects, and whether Institutional Review Board (IRB) approvals (or an equivalent approval/review based on the requirements of your country or institution) were obtained?

    Answer: [NA]

Justification: Our research does not involve human subjects, participants, or human data collection of any kind. All experiments use standard machine learning benchmark datasets (CIFAR-10/100, ImageNet) and computational evaluations. Therefore, no IRB approval or equivalent was required for this work.

Guidelines:

- The answer NA means that the paper does not involve crowdsourcing nor research with human subjects.
- Depending on the country in which research is conducted, IRB approval (or equivalent) may be required for any human subjects research. If you obtained IRB approval, you should clearly state this in the paper.
- We recognize that the procedures for this may vary significantly between institutions and locations, and we expect authors to adhere to the NeurIPS Code of Ethics and the guidelines for their institution.
- For initial submissions, do not include any information that would break anonymity (if applicable), such as the institution conducting the review.

16. **Declaration of LLM usage**

Question: Does the paper describe the usage of LLMs if it is an important, original, or non-standard component of the core methods in this research? Note that if the LLM is used only for writing, editing, or formatting purposes and does not impact the core methodology, scientific rigorousness, or originality of the research, declaration is not required.

Answer: [NA]

Justification: Our research on Variational Supervised Contrastive Learning (VarCon) does not involve Large Language Models (LLMs) in any capacity as part of our methodology, experiments, or analysis. Our work focuses on representation learning for image classification through a variational approach to contrastive learning, without any LLM components.

Guidelines:

- The answer NA means that the core method development in this research does not involve LLMs as any important, original, or non-standard components.
- Please refer to our LLM policy (`https://neurips.cc/Conferences/2025/LLM`) for what should or should not be described.

# Supplementary Material

## A Discussion

### A.1 Limitation

While Variational Supervised Contrastive Learning (VarCon) demonstrates significant improvements over existing methods, we acknowledge several limitations. First, our approach relies on supervised learning, which limits its applicability where labeled data is scarce or expensive to obtain. Second, although VarCon reduces comparison complexity from quadratic to nearly linear through class-level centroids, computing these centroids still introduces overhead for datasets with numerous classes. Third, cross-domain adaptation may require recalibration of our confidence-adaptive temperature mechanism when facing substantial distribution shifts between source and target domains.

Future work could address these limitations by: 1) extending VarCon to self-supervised settings by leveraging augmentation-based positive pairs with a variational formulation of latent pseudo-labels, which would maintain our probabilistic framework while removing the need for explicit supervision; 2) exploring amortized computation of class centroids through memory banks or representative embedding techniques to reduce computational overhead for large-scale applications; and 3) incorporating domain adaptation techniques such as distribution alignment constraints in the variational objective to improve cross-domain robustness. The core variational formulation of VarCon is flexible enough to accommodate these extensions with relatively modest modifications to the underlying framework.

Despite these limitations, VarCon successfully addresses a fundamental challenge in representation learning by providing a principled probabilistic interpretation of contrastive objectives while simultaneously improving empirical performance across diverse benchmarks. Our approach offers a new theoretical perspective that bridges the gap between generative and distinguishing paradigms, opening promising directions for developing more robust, interpretable, and efficient representation learning methods.

### A.2 Broader Impact

VarCon offers significant potential for broader impact across multiple domains of deep learning research and applications. By establishing a principled probabilistic foundation for contrastive learning and demonstrating substantial empirical improvements, our work contributes to several important areas:

From a scientific perspective, VarCon bridges the theoretical gap between generative and contrastive approaches to representation learning. This unification provides researchers with a more coherent understanding of how different learning paradigms relate to one another, potentially accelerating the development of hybrid approaches that leverage the strengths of multiple frameworks. The explicit modeling of uncertainty through our confidence-adaptive temperature mechanism also contributes to the growing body of work on uncertainty quantification in deep learning, which remains a critical challenge for reliable AI systems.

From an applications standpoint, VarCon's improved performance and robustness make it particularly valuable for domains where high-quality representations are essential. In medical imaging, more distinctive features could enhance diagnostic accuracy and treatment planning. In computer vision for autonomous vehicles, better representations may improve object detection and scene understanding under varying conditions. For scientific applications like protein structure prediction or molecular property estimation, our approach's ability to capture fine-grained semantic relationships could lead to meaningful discoveries.

The environmental impact of our approach is noteworthy. VarCon's reduced dependence on large batch sizes and faster convergence translate to lower computational requirements and energy consumption compared to existing contrastive methods. As the carbon footprint of machine learning research grows increasingly concerning, techniques that maintain or improve performance while requiring fewer computational resources represent an important direction for sustainability.

Our method's enhanced interpretability through uncertainty modeling provides a foundation for more transparent embedding learning. By explicitly quantifying confidence in learned representations,

VarCon could help identify potential failure modes or biases in downstream applications, particularly important as deep learning systems increasingly influence high-stakes decisions.

In summary, VarCon represents not only a technical advancement in representation learning but also a step toward more principled, efficient, and transparent deep learning systems, which can be responsibly deployed across a wide range of applications.

# B Experimental Analysis and Additional Results

## B.1 Hyperparameter Settings

In this section, we provide additional hyperparameter settings used in our VarCon implementation.

**Optimization Details** For all experiments, we used SGD with momentum 0.9 and weight decay $1 \times 10^{-4}$. We employed a cosine learning rate schedule with initial learning rate 0.05, and for experiments with batch sizes greater than 256, we incorporated a 10-epoch warm-up phase. For ImageNet experiments, we used the LARS optimizer to maintain training stability at large batch sizes. We utilized mixed-precision training (AMP) with gradient scaling to improve computational efficiency while maintaining numerical stability. All experiments were conducted on 8 NVIDIA A100 GPUs. Training for 350 epochs on ImageNet required approximately 54 hours.

**Data Augmentation Strategies** Data augmentation plays a crucial role in contrastive learning, as it defines the invariances that the representation should capture. We experimented with three augmentation strategies: SimAugment (random cropping, flipping, color jitter, and grayscale conversion), AutoAugment (which uses reinforcement learning to discover optimal transformation policies), and StackedRandAugment (which applies multiple random transformations sequentially). While previous contrastive learning methods often require strong augmentations to achieve competitive performance, VarCon demonstrates superior results even with simpler augmentation strategies. This is particularly significant because finding appropriate augmentation policies is typically domain-specific and time-consuming: augmentations that work well for images may not transfer to other modalities such as text or audio. VarCon's reduced dependency on aggressive augmentation makes it more adaptable across different domains and reduces the need for extensive hyperparameter tuning.

**Positive and Negative Sample Definition** Different contrastive learning frameworks define positive and negative samples distinctively, which significantly impacts their learning dynamics. In SimCLR, positive pairs consist solely of different augmented views of the same instance, while all other instances in the batch serve as negatives, regardless of their semantic similarity. SupCon extends this definition by considering samples from the same class as positives, leveraging label information to create semantically meaningful groupings. Our VarCon framework further refines this approach by encoding these relationships through a probabilistic lens. Rather than relying on explicit pairwise comparisons, VarCon infers class-conditional likelihoods and minimizes the KL divergence between the model's posterior and an adaptive target distribution. This formulation naturally handles both same-instance positives (through augmented views) and same-class positives (through class centroids) while maintaining appropriate separation from samples of different classes. The key distinction is that VarCon's confidence-adaptive temperature mechanism dynamically adjusts the "strength" of these positive relationships based on classification difficulty, providing fine-grained control over the embedding space organization. This probabilistic treatment of sample relationships contributes significantly to VarCon's superior performance and robustness across various experimental settings.

These implementation details contribute significantly to VarCon's performance and stability across different datasets and architectures, highlighting the importance of careful hyperparameter selection in representation learning systems.

## B.2 Extended Evaluations

To further validate the robustness of our proposed approach, we evaluated VarCon against leading self-supervised and supervised methods on the ImageNet-ReaL, which addresses the inherent limitations of the original ImageNet validation set by providing higher-quality multi-label annotations. Table 5 presents these results.

Table 5: Classification performance comparison on ImageNet-ReaL. We report Top-1 accuracy (%) (mean $\pm$ standard error) for VarCon versus state-of-the-art self-supervised and supervised methods. All models utilize the ResNet-50 architecture for a fair comparison. Best scores are highlighted in blue , second-best in green .

| Category | Method | Dataset |
|---|---|---|
| | | ImageNet-ReaL |
| | | Top-1 $\uparrow$ |
| Self-supervised | SimCLR | $75.30_{\pm 0.05}$ |
| | MoCo V2 | $78.22_{\pm 0.04}$ |
| | BYOL | $81.10_{\pm 0.07}$ |
| | SwAV | $81.56_{\pm 0.04}$ |
| | VicReg | $79.45_{\pm 0.07}$ |
| | Barlow Twins | $80.09_{\pm 0.03}$ |
| Supervised | Cross-Entropy | $83.47_{\pm 0.06}$ |
| | SupCon | $83.87_{\pm 0.04}$ |
| | **VarCon (Ours)** | $\mathbf{84.12}_{\pm 0.04}$ |

On this more rigorous dataset, VarCon achieves 84.12% Top-1 accuracy, outperforming all baseline methods by a statistically significant margin. The performance gap between VarCon and the strongest self-supervised method (SwAV at 81.56%) is significant, showing a solid 2.56% improvement, suggesting that our probabilistic formulation effectively leverages label information to learn more distinctive representations that better align with human perception of image content.

Even within the supervised category, VarCon demonstrates clear advantages over conventional cross-entropy (83.47%) and the previous state-of-the-art SupCon approach (83.87%). This improvement is especially meaningful on ImageNet-ReaL, where evaluation more accurately reflects model performance on actual visual content rather than potentially noisy single-label annotations. The consistent performance gains across both traditional ImageNet and ImageNet-ReaL datasets demonstrate that VarCon's improvements are not merely artifacts of label noise but represent genuine advances in representation quality. In addition, the reduced standard errors ($\pm 0.04$) compared to other methods indicate that VarCon's probabilistic approach not only improves accuracy but also yields more stable predictions across evaluation runs. This enhanced stability, combined with superior accuracy, highlights the effectiveness of our variational formulation in capturing the underlying structure of visual data while maintaining robust decision boundaries.

### B.3 Embedding Space Evolution Analysis

To investigate how VarCon shapes the embedding space during training, the progressive evolution of learned representations is visualized using t-SNE (perplexity 50) on ImageNet validation set. Models are trained for 200 epochs with checkpoints saved at epochs 50, 100, 150, 200. Figure 5 reveals systematic refinement of semantic organization throughout training.

The visualization demonstrates clear progression from loosely organized clusters at epoch 50 (KNN classifier: 52.74%) to highly structured, well-separated semantic groups by epoch 200 (79.11%). The most significant improvement occurs between epochs 100-150, where the KNN classifier performance jumps from 59.57% to 71.12%, corresponding to the convergence of our ELBO-derived loss function. During this phase, the KL divergence term effectively aligns the auxiliary posterior with the model's class posterior while ensuring correct class assignments. By epoch 200, VarCon achieves optimal embedding organization with distinct, compact clusters and clear decision boundaries. We employ KNN classification to directly evaluate the quality of pretrained representations without introducing additional trainable parameters that could mask the intrinsic separability of the learned embeddings. The high KNN classifier performance (79.11%) demonstrates that our confidence-adaptive temperature mechanism successfully provides fine-grained control over intra-class dispersion, creating

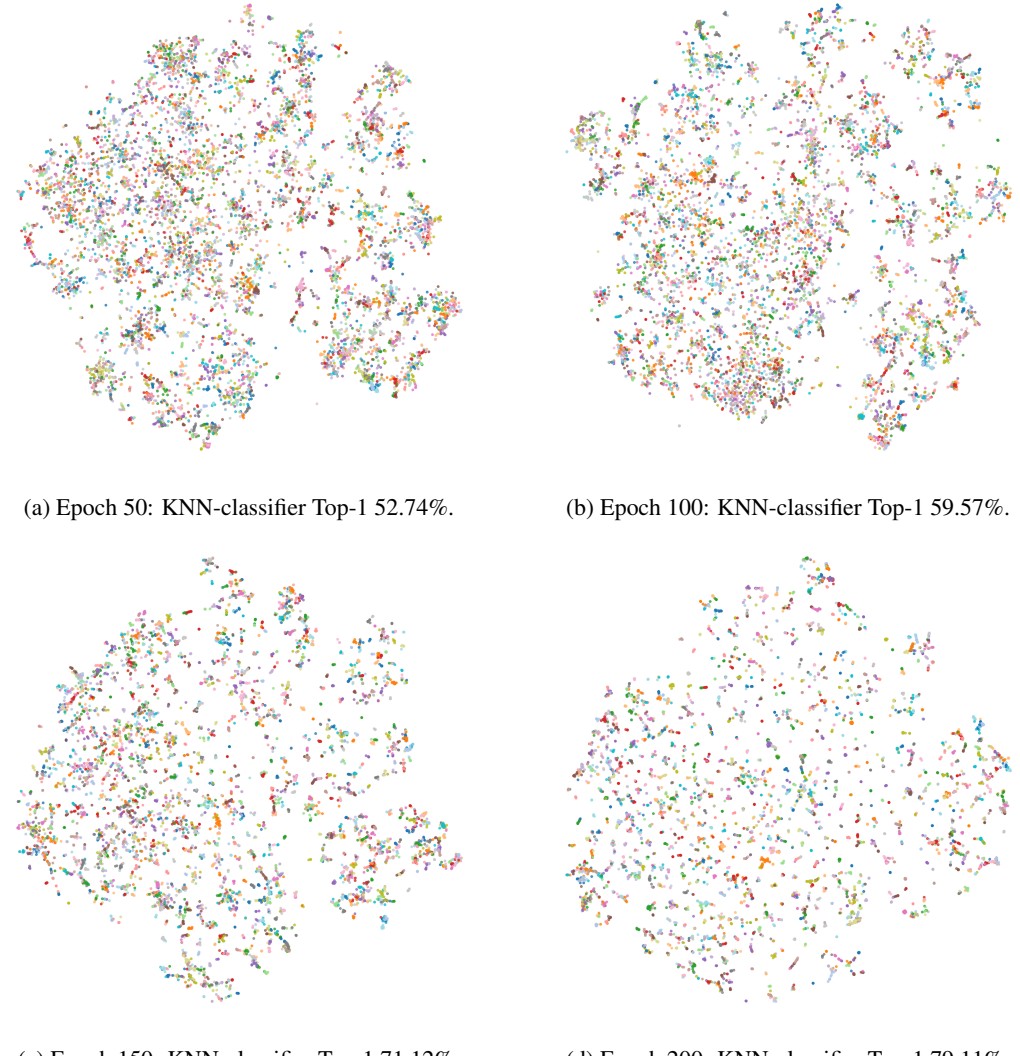

(a) Epoch 50: KNN-classifier Top-1 52.74%.

(b) Epoch 100: KNN-classifier Top-1 59.57%.

(c) Epoch 150: KNN-classifier Top-1 71.12%.

(d) Epoch 200: KNN-classifier Top-1 79.11%.

Figure 5: Progressive evolution of VarCon embedding space visualization through t-SNE [10] during training on ImageNet validation set. Our variational formulation demonstrates systematic improvement in semantic organization, with KNN-classifier accuracy increasing from 52.74% at epoch 50 to 79.11% at epoch 200 as clusters become increasingly well-separated and semantically coherent. The confidence-adaptive temperature mechanism enables fine-grained control over intra-class dispersion, resulting in embedding spaces with clear decision boundaries and hierarchical semantic structure that facilitate effective nearest-neighbor classification without additional parameterized classifiers.

embedding spaces where semantic similarity corresponds to geometric proximity and enabling effective classification through nearest-neighbor search.

The superior embedding quality achieved by VarCon stems from three synergistic mechanisms that fundamentally improve upon conventional contrastive learning. As demonstrated in Figure 6, VarCon achieves 79.11% KNN classifier performance in just 200 epochs, outperforming SupCon's 78.53% accuracy obtained after 350 epochs of training. First, by replacing exhaustive pairwise comparisons with class-level centroids, each sample directly learns to align with its corresponding class center, enabling more efficient identification of cluster centroids and reducing the quadratic computational complexity inherent in traditional contrastive methods. Second, our variational inference formulation preserves the possibility of inter-class linkages during the learning process, allowing the model to maintain nuanced relationships between semantically related classes rather than forcing rigid

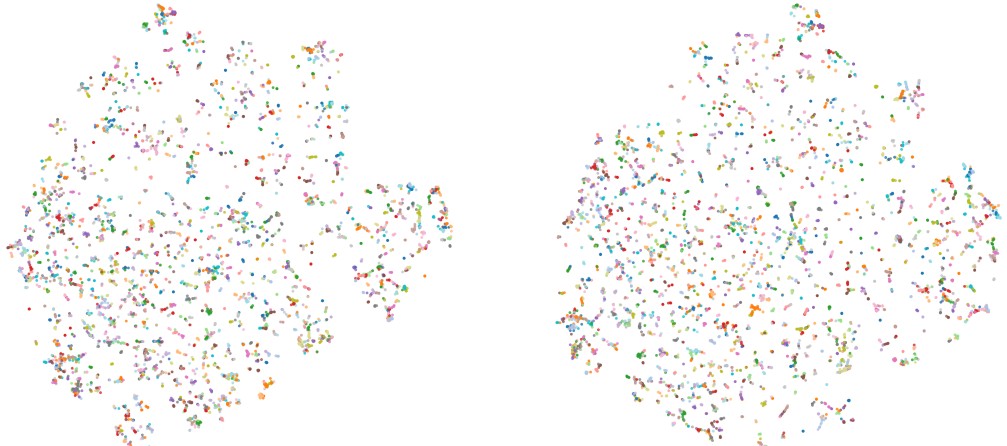

(a) SupCon (350 epochs): KNN-classifier Top-1 78.53%.

(b) VarCon (200 epochs): KNN-classifier Top-1 79.11%.

Figure 6: Embedding space comparison between SupCon and VarCon through t-SNE visualization on ImageNet validation set. Despite training for significantly fewer epochs (200 vs. 350), VarCon achieves superior KNN classifier performance (79.11% vs. 78.53%) and demonstrates clearer cluster separation. This comparison validates our findings in Figure 2b that VarCon converges faster than SupCon while simultaneously learning higher-quality representations with better-defined decision boundaries. The variational formulation enables more efficient optimization dynamics, achieving better semantic organization in substantially reduced training time.

Table 6: Hierarchical clustering evaluation on ImageNet embeddings. We perform Ward linkage clustering on the ImageNet validation set. All metrics: higher is better ($\uparrow$).

| Method | ARI | NMI | Homogeneity | Completeness | V-measure | Purity |
|---|---|---|---|---|---|---|
| SupCon | 0.613 | 0.888 | 0.886 | 0.890 | 0.888 | 0.755 |
| VarCon ($\epsilon = 0$) | 0.627 | 0.892 | 0.890 | 0.892 | 0.891 | 0.766 |
| **VarCon** | **0.634** | **0.895** | **0.893** | **0.896** | **0.895** | **0.774** |

separations. This probabilistic treatment enables the embedding space to capture gradual transitions and hierarchical relationships that are often lost in hard contrastive objectives. Third, the confidence-adaptive temperature mechanism provides dynamic regulation of learning intensity: for well-classified samples, it prevents overfitting by relaxing constraints, while simultaneously promoting continued learning on challenging examples by tightening supervision. This adaptive strategy ensures that computational resources are allocated efficiently, focusing learning capacity on samples that require additional refinement while maintaining stability for already well-separated instances.

## B.4 Hierarchical Clustering Results

To further investigate the semantic organization of learned representations, we perform hierarchical clustering analysis on the embedding space. We extract features from the entire ImageNet validation set and apply Ward linkage clustering, which iteratively merges clusters based on minimum variance criteria. This approach reveals the natural hierarchical structure present in the learned representations without imposing predefined category boundaries.

Table 6 presents comprehensive clustering quality metrics. VarCon consistently outperforms SupCon across all evaluation criteria, with progressive improvements observed through our ablation. The Adjusted Rand Index (ARI), which measures the similarity between cluster assignments and ground-truth labels while correcting for chance, increases from 0.613 (SupCon) to 0.627 (VarCon with $\epsilon = 0$) to 0.634 (VarCon). This progression suggests that the discovered clusters align increasingly closer with true semantic categories. Similarly, the purity metric, which quantifies the fraction of samples in

each cluster belonging to the most common class, improves from 0.755 to 0.766 to 0.774, indicating that individual clusters contain progressively fewer mixed-class samples.

The Normalized Mutual Information (NMI) score, measuring the mutual dependence between cluster assignments and true labels, increases from 0.888 to 0.892 to 0.895. This gain reflects a stronger statistical relationship between the discovered structure and semantic categories. The V-measure, which provides the harmonic mean of homogeneity and completeness, shows balanced improvements (0.888 to 0.891 to 0.895). Specifically, homogeneity, measuring whether each cluster contains only members of a single class, improves from 0.886 to 0.890 to 0.893, while completeness, measuring whether all members of a class are assigned to the same cluster, increases from 0.890 to 0.892 to 0.896. These metric improvements align with our theoretical framework. Notably, VarCon with fixed temperature ($\epsilon$) already outperforms SupCon, demonstrating that the ELBO-derived loss itself enhances representation quality. The full VarCon with confidence-adaptive temperature mechanism further improves performance by applying stronger supervision on difficult samples while relaxing constraints on well-classified examples. This adaptive behavior, combined with the KL divergence term that prevents distributional collapse, appears to preserve more nuanced semantic relationships during training. The clustering analysis thus provides empirical support for our variational formulation's ability to maintain meaningful structure in the representation space, with both components contributing to the superior performance.

## C   Theoretical Analysis

### C.1   Gradient Derivation w.r.t. Embedding $z$

To understand the training dynamics of VarCon, we derive the gradient of the VarCon loss with respect to the $\ell_2$-normalized embedding $z \in \mathbb{R}^d$. Our analysis reveals how the confidence-adaptive temperature mechanism influences learning through gradient modulation.

Recall that our VarCon loss is:

$$\mathcal{L}_{\text{VarCon}} = D_{\text{KL}}\big(q_\phi(r' \mid z) \,\|\, p_\theta(r' \mid z)\big) - \log p_\theta(r \mid z), \tag{14}$$

where $r$ is the ground-truth class, $q_\phi$ is the confidence-adaptive target distribution, and $p_\theta$ is the model's posterior distribution.

The gradient decomposes into two terms:

$$\frac{\partial \mathcal{L}_{\text{VarCon}}}{\partial z} = \underbrace{\frac{\partial D_{\text{KL}}}{\partial z}}_{\text{KL divergence term}} - \underbrace{\frac{\partial \log p_\theta(r \mid z)}{\partial z}}_{\text{Log-posterior term}}. \tag{15}$$

**1. Gradient of the KL Divergence.**   The KL divergence between $q_\phi$ and $p_\theta$ is:

$$D_{\text{KL}}\big(q_\phi \,\|\, p_\theta\big) = \sum_{r'=1}^{C} q_\phi(r' \mid z) \log \frac{q_\phi(r' \mid z)}{p_\theta(r' \mid z)}. \tag{16}$$

To compute its gradient in Eq. (16), we apply the product rule to each term:

$$\frac{\partial}{\partial z}\left[q_\phi(r' \mid z) \log \frac{q_\phi(r' \mid z)}{p_\theta(r' \mid z)}\right] = \frac{\partial q_\phi(r' \mid z)}{\partial z} \log \frac{q_\phi(r' \mid z)}{p_\theta(r' \mid z)}$$
$$+ q_\phi(r' \mid z) \frac{\partial}{\partial z} \log \frac{q_\phi(r' \mid z)}{p_\theta(r' \mid z)}. \tag{17}$$

For the second term in Eq. (17), we have:

$$q_\phi(r' \mid z)\frac{\partial}{\partial z} \log \frac{q_\phi(r' \mid z)}{p_\theta(r' \mid z)} = q_\phi(r' \mid z)\left[\frac{1}{q_\phi(r' \mid z)}\frac{\partial q_\phi(r' \mid z)}{\partial z} - \frac{1}{p_\theta(r' \mid z)}\frac{\partial p_\theta(r' \mid z)}{\partial z}\right]$$
$$= \frac{\partial q_\phi(r' \mid z)}{\partial z} - \frac{q_\phi(r' \mid z)}{p_\theta(r' \mid z)}\frac{\partial p_\theta(r' \mid z)}{\partial z}. \tag{18}$$

Therefore, the complete gradient of each term in Eq. (16) is:

$$\frac{\partial}{\partial z}\left[q_\phi(r' \mid z) \log \frac{q_\phi(r' \mid z)}{p_\theta(r' \mid z)}\right] = \frac{\partial q_\phi(r' \mid z)}{\partial z}\left[\log \frac{q_\phi(r' \mid z)}{p_\theta(r' \mid z)} + 1\right]$$
$$- \frac{q_\phi(r' \mid z)}{p_\theta(r' \mid z)}\frac{\partial p_\theta(r' \mid z)}{\partial z}. \tag{19}$$

Since $\sum_{r'} q_\phi(r' \mid z) = 1$, we have the constraint $\sum_{r'} \frac{\partial q_\phi(r'\mid z)}{\partial z} = 0$. Thus, when summing over all classes:

$$\frac{\partial D_{\mathrm{KL}}}{\partial z} = \sum_{r'=1}^{C} \frac{\partial q_\phi(r' \mid z)}{\partial z}\left[\log \frac{q_\phi(r' \mid z)}{p_\theta(r' \mid z)} + 1\right] - \sum_{r'=1}^{C} \frac{q_\phi(r' \mid z)}{p_\theta(r' \mid z)}\frac{\partial p_\theta(r' \mid z)}{\partial z}$$

$$= \sum_{r'=1}^{C} \frac{\partial q_\phi(r' \mid z)}{\partial z} \log \frac{q_\phi(r' \mid z)}{p_\theta(r' \mid z)} + \underbrace{\sum_{r'=1}^{C} \frac{\partial q_\phi(r' \mid z)}{\partial z}}_{=0} - \sum_{r'=1}^{C} \frac{q_\phi(r' \mid z)}{p_\theta(r' \mid z)}\frac{\partial p_\theta(r' \mid z)}{\partial z}$$

$$= \sum_{r'=1}^{C} \left[\frac{\partial q_\phi(r' \mid z)}{\partial z} \log \frac{q_\phi(r' \mid z)}{p_\theta(r' \mid z)} - \frac{q_\phi(r' \mid z)}{p_\theta(r' \mid z)}\frac{\partial p_\theta(r' \mid z)}{\partial z}\right]. \tag{20}$$

**Computing $\partial q_\phi(r' \mid z)/\partial z$.** The target distribution $q_\phi$ depends on $z$ through the confidence-adaptive temperature:

$$\tau_2(z) = (\tau_1 - \epsilon) + 2\epsilon p_\theta(r \mid z), \tag{21}$$

where $\epsilon$ is a learnable parameter controlling adaptation strength.

From our formulation, $q_\phi$ has the closed form:

$$q_\phi(r' \mid z) = \begin{cases} \frac{\exp(1/\tau_2)}{C-1+\exp(1/\tau_2)}, & \text{if } r' = r \\ \frac{1}{C-1+\exp(1/\tau_2)}, & \text{if } r' \neq r \end{cases} \tag{22}$$

For the ground-truth class $r' = r$:

$$\frac{\partial q_\phi(r \mid z)}{\partial \tau_2} = -\frac{(C-1)\exp(1/\tau_2)}{\tau_2^2[C-1+\exp(1/\tau_2)]^2}. \tag{23}$$

For other classes $r' \neq r$:

$$\frac{\partial q_\phi(r' \mid z)}{\partial \tau_2} = \frac{\exp(1/\tau_2)}{\tau_2^2[C-1+\exp(1/\tau_2)]^2}. \tag{24}$$

By the chain rule:

$$\frac{\partial q_\phi(r' \mid z)}{\partial z} = \frac{\partial q_\phi(r' \mid z)}{\partial \tau_2} \cdot \frac{\partial \tau_2}{\partial z} = \frac{\partial q_\phi(r' \mid z)}{\partial \tau_2} \cdot 2\epsilon \frac{\partial p_\theta(r \mid z)}{\partial z}. \tag{25}$$

**Computing $\partial p_\theta(r' \mid z)/\partial z$.** The model's posterior follows a softmax distribution:

$$p_\theta(r' \mid z) = \frac{\exp(z^\top w_{r'}/\tau_1)}{\sum_{k=1}^{C} \exp(z^\top w_k/\tau_1)}. \tag{26}$$

**Remark on Class Centroid Computation.** In our implementation, we compute class centroids as the normalized average of embeddings:

$$w_r = \frac{\bar{z}_r}{\|\bar{z}_r\|_2}, \quad \text{where} \quad \bar{z}_r = \frac{1}{|B_r|}\sum_{i \in B_r} z_i, \tag{27}$$

and $B_r = \{i : r_i = r, i \in B\}$ denotes the set of samples with class $r$ in the mini-batch. We treat these centroids as constants during backpropagation (i.e., we detach them from the computational graph). This design choice:

- Simplifies gradient computation and improves training efficiency

- Avoids cyclic dependencies that could lead to training instabilities

- Aligns with the interpretation of centroids as fixed reference points representing the current state of each class

Therefore, when computing $\partial \mathcal{L}_{\text{VarCon}}/\partial \boldsymbol{z}$, we do not propagate gradients through $\boldsymbol{w}_r$, treating them as constants in the following derivations.

Its gradient is:

$$\frac{\partial p_\theta(r' \mid \boldsymbol{z})}{\partial \boldsymbol{z}} = \frac{p_\theta(r' \mid \boldsymbol{z})}{\tau_1} \left[ \boldsymbol{w}_{r'} - \mathbb{E}_{p_\theta}[\boldsymbol{w}] \right], \tag{28}$$

where $\mathbb{E}_{p_\theta}[\boldsymbol{w}] = \sum_{k=1}^{C} p_\theta(k \mid \boldsymbol{z}) \boldsymbol{w}_k$ is the expected class centroid under the current model distribution.

**2. Gradient of the Log-Posterior.**    From the log-posterior definition:

$$\log p_\theta(r \mid \boldsymbol{z}) = \frac{\boldsymbol{z}^\top \boldsymbol{w}_r}{\tau_1} - \log \sum_{r'=1}^{C} \exp\left( \frac{\boldsymbol{z}^\top \boldsymbol{w}_{r'}}{\tau_1} \right). \tag{29}$$

Taking the gradient of Eq. (29):

$$\frac{\partial \log p_\theta(r \mid \boldsymbol{z})}{\partial \boldsymbol{z}} = \frac{1}{\tau_1} \left[ \boldsymbol{w}_r - \mathbb{E}_{p_\theta}[\boldsymbol{w}] \right]. \tag{30}$$

**Complete Gradient and Interpretation.**    Combining both Eqs. (20) and (30), the complete gradient is:

$$\frac{\partial \mathcal{L}_{\text{VarCon}}}{\partial \boldsymbol{z}} = \sum_{r'=1}^{C} \left[ \underbrace{\frac{\partial q_\phi(r' \mid \boldsymbol{z})}{\partial \boldsymbol{z}} \log \frac{q_\phi(r' \mid \boldsymbol{z})}{p_\theta(r' \mid \boldsymbol{z})}}_{\text{Distribution alignment}} - \underbrace{\frac{q_\phi(r' \mid \boldsymbol{z})}{p_\theta(r' \mid \boldsymbol{z})} \frac{\partial p_\theta(r' \mid \boldsymbol{z})}{\partial \boldsymbol{z}}}_{\text{Weighted gradient}} \right]$$

$$- \underbrace{\frac{1}{\tau_1} \left[ \boldsymbol{w}_r - \mathbb{E}_{p_\theta}[\boldsymbol{w}] \right]}_{\text{Centroid attraction}}. \tag{31}$$

Since embeddings are $\ell_2$-normalized ($\|\boldsymbol{z}\|_2 = 1$), the effective gradient must be projected onto the tangent space of the unit sphere:

$$\left. \frac{\partial \mathcal{L}_{\text{VarCon}}}{\partial \boldsymbol{z}} \right|_{\text{effective}} = \left( \mathbf{I} - \boldsymbol{z}\boldsymbol{z}^\top \right) \frac{\partial \mathcal{L}_{\text{VarCon}}}{\partial \boldsymbol{z}}, \tag{32}$$

where $\mathbf{I} - \boldsymbol{z}\boldsymbol{z}^\top$ is the projection operator.

This gradient analysis reveals the dual mechanism of VarCon:

- The KL divergence term aligns the auxiliary distribution $q_\phi$ with the model posterior $p_\theta$ through confidence-adaptive weighting.

- The log-posterior term creates an attractive force toward the true class centroid $\boldsymbol{w}_r$ and a repulsive force from the expected centroid $\mathbb{E}_{p_\theta}[\boldsymbol{w}]$.

- The confidence-adaptive temperature $\tau_2(\boldsymbol{z})$ modulates the alignment strength based on classification confidence, providing sharper supervision for difficult samples.

### C.2    Gradient Derivation w.r.t. Adaptation Strength Parameter $\epsilon$

To understand how the adaptation strength parameter $\epsilon$ influences the training dynamics, we derive the gradient of the VarCon loss with respect to $\epsilon$. This analysis reveals how the confidence-adaptive mechanism automatically adjusts the strength of supervision based on sample difficulty.

Recall that $\epsilon$ appears in the confidence-adaptive temperature:

$$\tau_2(\mathbf{z}) = (\tau_1 - \epsilon) + 2\epsilon p_\theta(r \mid \mathbf{z}),$$

which in turn affects the target distribution $q_\phi$. The gradient of the VarCon loss with respect to $\epsilon$ is:

$$\frac{\partial \mathcal{L}_{\text{VarCon}}}{\partial \epsilon} = \frac{\partial D_{\text{KL}}}{\partial \epsilon} - \frac{\partial \log p_\theta(r \mid \mathbf{z})}{\partial \epsilon}. \tag{33}$$

We compute the instantaneous gradient with respect to $\epsilon$ while treating the encoder parameters $\theta$ as fixed during this computation. This is consistent with how gradients are computed in backpropagation, where each parameter's gradient is calculated independently. Under this assumption, $p_\theta(r \mid \mathbf{z})$ is treated as a constant with respect to $\epsilon$, yielding:

$$\left. \frac{\partial \log p_\theta(r \mid \mathbf{z})}{\partial \epsilon} \right|_{\theta \text{ fixed}} = 0. \tag{34}$$

Therefore, the gradient in Eq. (33) simplifies to:

$$\frac{\partial \mathcal{L}_{\text{VarCon}}}{\partial \epsilon} = \frac{\partial D_{\text{KL}}}{\partial \epsilon}. \tag{35}$$

**Computing the KL Divergence Gradient.** From the KL divergence definition in Eq. (16):

$$D_{\text{KL}} = \sum_{r'=1}^{C} q_\phi(r' \mid \mathbf{z}) \log \frac{q_\phi(r' \mid \mathbf{z})}{p_\theta(r' \mid \mathbf{z})},$$

we apply the product rule to compute its derivative with respect to $\epsilon$:

$$\frac{\partial D_{\text{KL}}}{\partial \epsilon} = \sum_{r'=1}^{C} \left[ \frac{\partial q_\phi(r' \mid \mathbf{z})}{\partial \epsilon} \log \frac{q_\phi(r' \mid \mathbf{z})}{p_\theta(r' \mid \mathbf{z})} + q_\phi(r' \mid \mathbf{z}) \frac{\partial}{\partial \epsilon} \log \frac{q_\phi(r' \mid \mathbf{z})}{p_\theta(r' \mid \mathbf{z})} \right]. \tag{36}$$

For the second term in Eq. (36), since $p_\theta(r' \mid \mathbf{z})$ is treated as constant with respect to $\epsilon$:

$$\frac{\partial}{\partial \epsilon} \log \frac{q_\phi(r' \mid \mathbf{z})}{p_\theta(r' \mid \mathbf{z})} = \frac{1}{q_\phi(r' \mid \mathbf{z})} \frac{\partial q_\phi(r' \mid \mathbf{z})}{\partial \epsilon}. \tag{37}$$

Substituting back in Eq. (36):

$$\begin{aligned}
\frac{\partial D_{\text{KL}}}{\partial \epsilon} &= \sum_{r'=1}^{C} \left[ \frac{\partial q_\phi(r' \mid \mathbf{z})}{\partial \epsilon} \log \frac{q_\phi(r' \mid \mathbf{z})}{p_\theta(r' \mid \mathbf{z})} + q_\phi(r' \mid \mathbf{z}) \cdot \frac{1}{q_\phi(r' \mid \mathbf{z})} \frac{\partial q_\phi(r' \mid \mathbf{z})}{\partial \epsilon} \right] \\
&= \sum_{r'=1}^{C} \frac{\partial q_\phi(r' \mid \mathbf{z})}{\partial \epsilon} \left[ \log \frac{q_\phi(r' \mid \mathbf{z})}{p_\theta(r' \mid \mathbf{z})} + 1 \right].
\end{aligned} \tag{38}$$

Since $\sum_{r'} q_\phi(r' \mid \mathbf{z}) = 1$, we have the constraint $\sum_{r'} \frac{\partial q_\phi(r' \mid \mathbf{z})}{\partial \epsilon} = 0$. Thus:

$$\begin{aligned}
\frac{\partial D_{\text{KL}}}{\partial \epsilon} &= \sum_{r'=1}^{C} \frac{\partial q_\phi(r' \mid \mathbf{z})}{\partial \epsilon} \left[ \log \frac{q_\phi(r' \mid \mathbf{z})}{p_\theta(r' \mid \mathbf{z})} + 1 \right] \\
&= \sum_{r'=1}^{C} \frac{\partial q_\phi(r' \mid \mathbf{z})}{\partial \epsilon} \log \frac{q_\phi(r' \mid \mathbf{z})}{p_\theta(r' \mid \mathbf{z})} + \underbrace{\sum_{r'=1}^{C} \frac{\partial q_\phi(r' \mid \mathbf{z})}{\partial \epsilon}}_{=0} \\
&= \sum_{r'=1}^{C} \frac{\partial q_\phi(r' \mid \mathbf{z})}{\partial \epsilon} \log \frac{q_\phi(r' \mid \mathbf{z})}{p_\theta(r' \mid \mathbf{z})}.
\end{aligned} \tag{39}$$

**Computing $\partial q_\phi(r' \mid z)/\partial\epsilon$.**   Using the chain rule:

$$\frac{\partial q_\phi(r' \mid z)}{\partial\epsilon} = \frac{\partial q_\phi(r' \mid z)}{\partial\tau_2} \cdot \frac{\partial\tau_2}{\partial\epsilon}. \tag{40}$$

First, we compute $\partial\tau_2/\partial\epsilon$:

$$\begin{aligned}
\frac{\partial\tau_2}{\partial\epsilon} &= \frac{\partial}{\partial\epsilon}\left[(\tau_1 - \epsilon) + 2\epsilon p_\theta(r \mid z)\right] \\
&= -1 + 2p_\theta(r \mid z) \\
&= 2p_\theta(r \mid z) - 1.
\end{aligned} \tag{41}$$

Note that $p_\theta(r \mid z)$ is treated as a constant with respect to $\epsilon$ under our instantaneous gradient assumption.

This derivative reveals a key insight:

- When $p_\theta(r \mid z) > 0.5$ (confident predictions): $\partial\tau_2/\partial\epsilon > 0$
- When $p_\theta(r \mid z) < 0.5$ (difficult samples): $\partial\tau_2/\partial\epsilon < 0$

The derivatives $\partial q_\phi(r' \mid z)/\partial\tau_2$ were computed in Section C.1. For the ground-truth class $r' = r$:

$$\frac{\partial q_\phi(r \mid z)}{\partial\tau_2} = -\frac{(C-1)\exp(1/\tau_2)}{\tau_2^2[C - 1 + \exp(1/\tau_2)]^2}, \tag{42}$$

and for other classes $r' \neq r$:

$$\frac{\partial q_\phi(r' \mid z)}{\partial\tau_2} = \frac{\exp(1/\tau_2)}{\tau_2^2[C - 1 + \exp(1/\tau_2)]^2}. \tag{43}$$

**Complete Gradient Expression.**   Combining all terms in Eqs. (39) and (41):

$$\begin{aligned}
\frac{\partial\mathcal{L}_{\text{VarCon}}}{\partial\epsilon} &= \sum_{r'=1}^{C} \frac{\partial q_\phi(r' \mid z)}{\partial\tau_2} \cdot \frac{\partial\tau_2}{\partial\epsilon} \cdot \log\frac{q_\phi(r' \mid z)}{p_\theta(r' \mid z)} \\
&= [2p_\theta(r \mid z) - 1]\sum_{r'=1}^{C} \frac{\partial q_\phi(r' \mid z)}{\partial\tau_2} \log\frac{q_\phi(r' \mid z)}{p_\theta(r' \mid z)}.
\end{aligned} \tag{44}$$

Substituting the specific derivatives in Eqs. (42) and (43):

$$\begin{aligned}
\frac{\partial\mathcal{L}_{\text{VarCon}}}{\partial\epsilon} = [2p_\theta(r \mid z) - 1] \times \Bigg\{ & \\
\sum_{r'\neq r} \frac{\exp(1/\tau_2)}{\tau_2^2[C - 1 + \exp(1/\tau_2)]^2} &\log\frac{q_\phi(r' \mid z)}{p_\theta(r' \mid z)} \\
-\frac{(C-1)\exp(1/\tau_2)}{\tau_2^2[C - 1 + \exp(1/\tau_2)]^2} &\log\frac{q_\phi(r \mid z)}{p_\theta(r \mid z)} \Bigg\}.
\end{aligned} \tag{45}$$

Factoring out common terms:

$$\begin{aligned}
\frac{\partial\mathcal{L}_{\text{VarCon}}}{\partial\epsilon} = {}& \frac{[2p_\theta(r \mid z) - 1]\exp(1/\tau_2)}{\tau_2^2[C - 1 + \exp(1/\tau_2)]^2} \\
& \times \left\{\sum_{r'\neq r} \log\frac{q_\phi(r' \mid z)}{p_\theta(r' \mid z)} - (C-1)\log\frac{q_\phi(r \mid z)}{p_\theta(r \mid z)}\right\}.
\end{aligned} \tag{46}$$

**Interpretation and Learning Dynamics.** This gradient reveals the self-regulating nature of the confidence-adaptive mechanism:

- **For confident samples** ($p_\theta(r \mid \boldsymbol{z}) > 0.5$): The factor $[2p_\theta(r \mid \boldsymbol{z}) - 1] > 0$. When $p_\theta(r \mid \boldsymbol{z})$ is high (confident prediction), $\tau_2$ approaches $\tau_1 + \epsilon$, making $q_\phi$ more uniform. The gradient direction is determined by the aggregate log-ratio term $\sum_{r' \neq r} \log \frac{q_\phi(r'|\boldsymbol{z})}{p_\theta(r'|\boldsymbol{z})} - (C-1) \log \frac{q_\phi(r|\boldsymbol{z})}{p_\theta(r|\boldsymbol{z})}$. This uniform $q_\phi$ typically satisfies $q_\phi(r \mid \boldsymbol{z}) < p_\theta(r \mid \boldsymbol{z})$ for the ground-truth class and $q_\phi(r' \mid \boldsymbol{z}) > p_\theta(r' \mid \boldsymbol{z})$ for other classes, making the aggregate term positive. Consequently, the gradient pushes $\epsilon$ to increase, which in turn increases $\tau_2$, making $q_\phi$ even more uniform and effectively reducing supervision strength for already well-classified samples. This self-reinforcing mechanism prevents overfitting on easy samples while maintaining stable learning dynamics.

- **For difficult samples** ($p_\theta(r \mid \boldsymbol{z}) < 0.5$): The factor $[2p_\theta(r \mid \boldsymbol{z}) - 1] < 0$. When the model is uncertain, $\tau_2$ approaches $\tau_1 - \epsilon$, creating a sharper distribution. The gradient behavior reverses, causing $\epsilon$ to adjust such that $\tau_2$ decreases. This reduction in $\epsilon$ leads to a smaller $\tau_2$, creating a sharper $q_\phi$ distribution that provides stronger supervision precisely where the model needs it most. The same aggregate log-ratio term that increases $\epsilon$ for confident samples now decreases it due to the negative sign factor. The magnitude of this effect scales with the degree of distributional mismatch, ensuring proportional adaptation to the model's uncertainty.

- **Dynamic equilibrium**: During training, $\epsilon$ and $\theta$ are updated jointly through gradient descent. While our analysis considers their instantaneous gradients separately, their co-evolution creates a dynamic equilibrium where $\epsilon$ continuously adapts to the current state of the model, automatically balancing exploration and exploitation throughout the learning process. As training progresses and the model's predictions align better with ground truth, the distributional differences driving the gradient naturally diminish, stabilizing $\epsilon$. Although $\epsilon$ is learnable and updated via gradient descent according to the derived gradients, we enforce hard bounds during optimization to ensure numerical stability. Specifically, after each gradient update, $\epsilon$ is clamped to a predefined range (e.g., $[0, 0.08]$ as illustrated in Figure 3b). This constraint ensures that the confidence-adaptive temperature $\tau_2(\boldsymbol{z}) = (\tau_1 - \epsilon) + 2\epsilon p_\theta(r \mid \boldsymbol{z})$ remains within valid bounds throughout training, preventing degenerate solutions while still allowing sufficient flexibility for adaptation.

This analysis confirms that the learnable parameter $\epsilon$ provides an elegant mechanism for adaptive supervision strength, contributing to VarCon's superior performance and faster convergence compared to fixed-temperature approaches.

# D Training Details

## D.1 Loss Convergence Analysis

To empirically validate VarCon's training efficiency, we analyze the loss trajectories in a controlled 50-epoch training experiment on ImageNet with ResNet-50 and an overall batch size of 4096 distributed across 8 A100 GPUs, as demonstrated in Figure 7. VarCon demonstrates dramatically faster loss convergence compared to SupCon: starting from comparable initial losses (VarCon: 9.997, SupCon: 11.142), VarCon's loss drops to 2.836 by epoch 10, which is a 71.6% reduction versus SupCon's 34.3% reduction to 7.311. This convergence advantage persists throughout training, with VarCon achieving a final loss of 1.138 at epoch 50, 75.1% lower than SupCon's 4.578. Crucially, both components of VarCon's objective decrease synergistically: the KL divergence term reduces from 4.978 to 0.538 (89.2% decrease) while the negative log-posterior term decreases from 5.019 to 0.599 (88.1% reduction), confirming that our variational formulation successfully balances distributional alignment with classification performance. Even with this abbreviated 50-epoch training schedule, significantly shorter than the standard 200-350 epochs, both methods achieve reasonable downstream classification accuracy (VarCon: 75.3%, SupCon: 74.5%), but VarCon's substantially lower final loss and steeper descent trajectory demonstrate more efficient optimization. This empirical evidence validates our theoretical framework: the ELBO-derived objective provides better gradient signals, while the

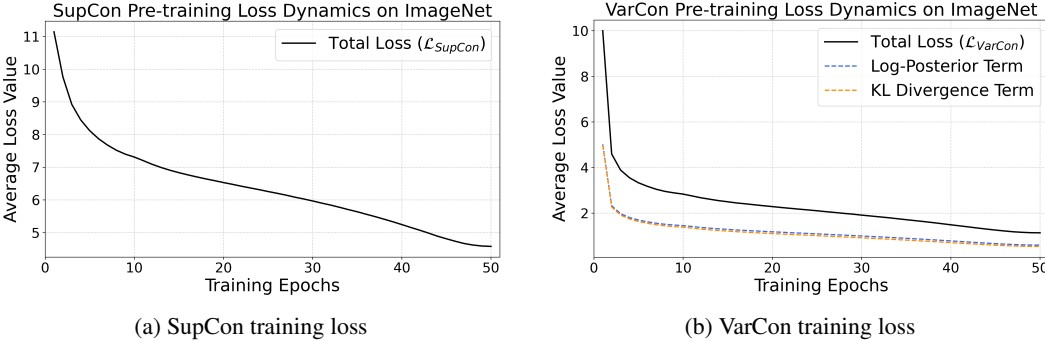

(a) SupCon training loss          (b) VarCon training loss

Figure 7: Training loss trajectories on ImageNet with ResNet-50 and batch size 4096 under a full 50-epoch training regime. (a) SupCon exhibits slow convergence, decreasing from 11.142 (epoch 1) to 4.578 (epoch 50). (b) VarCon demonstrates rapid convergence with the total loss dropping from 9.997 (epoch 1) to 1.138 (epoch 50), while both components decrease synergistically: the KL divergence term reduces from 4.978 (epoch 1) to 0.538 (epoch 50) and the negative log-posterior term from 5.019 (epoch 1) to 0.599 (epoch 50). This balanced reduction validates our variational formulation's effectiveness in jointly optimizing distributional alignment and class-specific representation learning.

Table 7: Distribution of adaptive temperature $\tau_2$ across ImageNet samples at key epochs during a full 50-epoch training with ResNet-50, $\epsilon = 0.02$, $\tau_1 = 0.1$, and batch size 4096. The temperature $\tau_2 \in [0.08, 0.12]$ modulates gradient strength, with lower values indicating stronger gradients for difficult samples. "25th Pct" and "75th Pct" represent the first quartile (25th percentile) and third quartile (75th percentile), respectively.

| Epoch | Mean | Std Dev | Min | 25th Pct | Median | 75th Pct | Max | $\epsilon$ |
|---|---|---|---|---|---|---|---|---|
| 10 | 0.09378 | 0.00961 | 0.08001 | 0.08550 | 0.09213 | 0.10110 | 0.11878 | 0.02 |
| 20 | 0.09714 | 0.01037 | 0.08000 | 0.08806 | 0.09685 | 0.10610 | 0.11873 | 0.02 |
| 30 | 0.09979 | 0.01078 | 0.08000 | 0.09063 | 0.10076 | 0.10935 | 0.11889 | 0.02 |
| 40 | 0.10326 | 0.01092 | 0.08001 | 0.09505 | 0.10589 | 0.11278 | 0.11916 | 0.02 |
| 50 | 0.10656 | 0.01059 | 0.08001 | 0.10040 | 0.11042 | 0.11520 | 0.11910 | 0.02 |

confidence-adaptive temperature mechanism accelerates learning without sacrificing representation quality.

### D.2  Quantitative Analysis of Adaptive Temperature Dynamics

To investigate the effectiveness of our confidence-adaptive temperature mechanism, we analyze the distribution of $\tau_2$ across all ImageNet samples during a 50-epoch training run with ResNet-50, fixed hyperparameters ($\epsilon = 0.02$, $\tau_1 = 0.1$) and batch size 4096, constraining $\tau_2 \in [0.08, 0.12]$. Table 7 and Figure 4 reveal systematic evolution of the temperature distribution as training progresses. At epoch 10, the mean $\tau_2 = 0.09378$ with the 25th percentile at 0.08550 indicates that over 25% of samples receive near-maximal gradient strength, while the relatively low 75th percentile of 0.10110 shows that even easier, more confident samples remain below the midpoint of the temperature range. The distribution shifts consistently across epochs, as visualized in Figure 4a: the mean increases by 13.6% (from 0.09378 to 0.10656), while more dramatic shifts occur in the quartiles—the 25th percentile rises by 17.4% (from 0.08550 to 0.10040) and the median increases by 19.9% (from 0.09213 to 0.11042), indicating accelerating confidence gains on the majority of samples. The minimum and maximum values follow divergent trajectories: while the minimum remains fixed at around 0.08001 throughout training (suggesting a persistent subset of challenging samples), Figure 4b reveals that the density of low-$\tau_2$ samples progressively decreases from epoch 10 to 50, demonstrating that hard samples are gradually pushed closer to their corresponding class centroids through embedding refinement. Meanwhile, the maximum stabilizes around 0.119, and the 75th percentile increases from 0.10110 to 0.11520 (14.0% growth), demonstrating that high-confidence samples progressively receive more relaxed gradients. The standard deviation exhibits a subtle but meaningful pattern, initially increasing from 0.00961 to 0.01092 (epochs 10-40) before slightly decreasing to 0.01059

Table 8: Distribution of adaptive temperature $\tau_2$ across ImageNet samples with learnable $\epsilon$ during a full 200-epoch training with ResNet-50, $\tau_1 = 0.1$, and batch size 4096. The temperature $\tau_2$ modulates gradient strength, with lower values indicating stronger gradients for difficult samples. The learnable $\epsilon$ parameter evolves dynamically during training.

| Epoch | Mean | Std Dev | Min | 25th Pct | Median | 75th Pct | Max | $\epsilon$ |
|---|---|---|---|---|---|---|---|---|
| 40 | 0.09365 | 0.00958 | 0.08151 | 0.08542 | 0.09205 | 0.10098 | 0.11845 | 0.0185 |
| 80 | 0.09728 | 0.01042 | 0.07950 | 0.08815 | 0.09698 | 0.10625 | 0.11885 | 0.0205 |
| 120 | 0.09995 | 0.01085 | 0.07850 | 0.09075 | 0.10090 | 0.10950 | 0.11902 | 0.0215 |
| 160 | 0.10310 | 0.01088 | 0.07921 | 0.09492 | 0.10575 | 0.11265 | 0.11895 | 0.0208 |
| 200 | 0.10642 | 0.01056 | 0.07981 | 0.10028 | 0.11030 | 0.11508 | 0.11887 | 0.0202 |

at epoch 50, suggesting initial divergence as samples differentiate followed by stabilization. This nuanced evolution, with faster increases in upper quantiles compared to lower ones, illustrates that our confidence-weighted formulation successfully implements differential treatment: maintaining strong gradients for persistently difficult samples while progressively relaxing constraints on well-learned examples, all emerging naturally without manual scheduling. This emergent behavior represents a fundamental advance over existing approaches that either apply uniform treatment to all samples or rely on handcrafted scheduling rules. The emergence of this heterogeneous learning dynamic, where gradient strength naturally adapts to each sample's learning progress, demonstrates how our variational framework translates theoretical principles into practical benefits. Rather than requiring explicit sample weighting or curriculum strategies, the adaptive behavior arises from the interaction between the KL divergence term and the confidence-based temperature mechanism, providing a principled solution rather than manual scheduling approaches.

To examine the adaptive capacity of our framework with learnable parameters, we extend our analysis to a full 200-epoch training schedule. Table 8 demonstrates the co-evolution of $\tau_2$ distribution and the learnable $\epsilon$ parameter. The $\epsilon$ parameter exhibits non-monotonic dynamics: starting at 0.0185 (epoch 40), increasing to a peak of 0.0215 (epoch 120), then gradually decreasing to 0.0202 (epoch 200). This trajectory reflects the model's evolving calibration requirements across different training phases. The $\tau_2$ distribution under learnable $\epsilon$ shows similar overall trends to the fixed-$\epsilon$ setting but with notable distinctions. The mean $\tau_2$ increases from 0.09365 (epoch 40) to 0.10642 (epoch 200), representing 13.6% growth comparable to the fixed case. However, the distribution exhibits greater dynamic range: minimum values decrease from 0.08151 (epoch 40) to 0.07981 (epoch 200), enabling more aggressive gradient modulation for the hardest samples. The 25th percentile increases by 17.3% (0.08542 to 0.10028) and the median by 19.9% (0.09205 to 0.11030), demonstrating consistent confidence development across the sample population. The interplay between $\epsilon$ and $\tau_2$ reveals sophisticated adaptation patterns. During epochs 40-120, $\epsilon$ increases from 0.0185 to 0.0215, expanding the temperature range to facilitate stronger differentiation between easy and hard samples. As training progresses (epochs 120-200), $\epsilon$ decreases to 0.0202 while maintaining high mean $\tau_2$, indicating the model has established robust class boundaries and requires less aggressive temperature modulation. The standard deviation remains stable (0.00958 to 0.01056) despite evolving $\epsilon$, demonstrating that the learnable mechanism maintains consistent relative differentiation across samples while adjusting absolute scale. This self-regulated behavior emerges without explicit scheduling, validating our variational framework's capacity for autonomous adaptation to training dynamics.

### D.3 Memory Efficiency Analysis

Beyond training efficiency, we analyze the memory footprint of VarCon compared to SupCon during training. Figure 8 presents peak GPU memory consumption per-device (specifically Rank 0) over 50 training epochs on ImageNet with ResNet-50 and per-device batch size 512. VarCon demonstrates consistent memory efficiency advantages: after the initial epoch, VarCon maintains a steady peak memory usage of 43.09 GB per GPU compared to SupCon's 43.25 GB, representing a 0.16 GB (0.37%) reduction per device. While this difference appears modest, it becomes significant in multi-GPU training scenarios where even small per-device savings enable larger effective batch sizes or accommodate additional model capacity within fixed memory constraints.

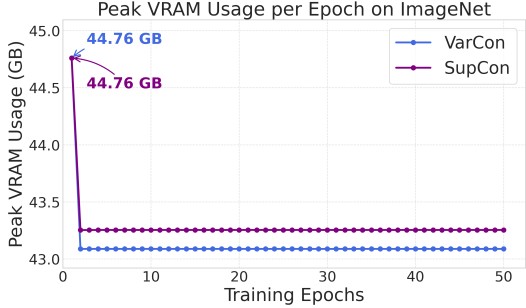

Figure 8: Per-GPU peak memory usage comparison between VarCon and SupCon during a full 50-epoch ImageNet training with ResNet-50. Experiments conducted with total batch size 4096 distributed across 8 A100 GPUs (per-GPU batch size 512), with VarCon using $\epsilon = 0.02$ and $\tau_1 = 0.1$. VarCon consistently requires 0.16 GB less memory per device after the first epoch, maintaining 43.09 GB versus SupCon's 43.25 GB throughout training.

Table 9: Comparison of adaptive regularization methods across four key dimensions. Dynamic Target: modifies target distribution based on sample or model state; Dynamic Gradient: adjusts gradient magnitude during training; Sample-Specific: per-sample adaptive treatment; Self-Adaptive: automatic confidence-based updates without manual scheduling.

| Method | Dynamic Target | Dynamic Gradient | Sample-Specific | Self-Adaptive |
|---|---|---|---|---|
| Label Smoothing [57] | ✗ | ✗ | ✗ | ✗ |
| Mixup [67] | ✗ | ✗ | ✗ | ✗ |
| AdaFocal [27] | ✗ | ✓ | ✓ | ✓ |
| Dual Focal Loss [58] | ✗ | ✓ | ✓ | ✗ |
| Local Adaptive LS [4] | ✓ | ✗ | ✓ | ✗ |
| LoT Regularization [36] | ✓ | ✓ | ✓ | ✗ |
| **VarCon (Ours)** | ✓ | ✓ | ✓ | ✓ |

The memory savings arise from VarCon's computational structure: by computing class centroids dynamically rather than exhaustive pairwise comparisons, VarCon reduces the memory required for intermediate similarity matrices from $O(B^2)$ to $O(B \cdot C_d)$, where $B$ denotes the per-device batch size and $C_d$ represents the number of unique classes per device. Importantly, $C_d$ is bounded by $B$ (i.e., $C_d \leq B$), as each sample belongs to exactly one class. This bound ensures that VarCon's memory complexity remains strictly lower than SupCon's quadratic requirement. Furthermore, the stable memory consumption across epochs indicates that the confidence-adaptive mechanism introduces negligible overhead despite its dynamic nature. These efficiency gains, combined with VarCon's faster convergence and superior accuracy, establish it as a more resource-efficient approach for contrastive learning at scale.

### D.4 Adaptive Temperature versus Existing Regularization Methods

Traditional methods like label smoothing [57] and mixup [67] apply uniform regularization to all samples, while focal loss variants such as AdaFocal [27] and Dual Focal Loss [58] adjust gradient magnitudes based on sample difficulty. Recent approaches achieve sample-specific adaptation: Local Adaptive Label Smoothing [4] modifies smoothing based on neighborhood structure, while Learning from Teaching regularization [36] combines target and gradient modifications through teacher-student dynamics. As summarized in Table 9, these methods primarily optimize either target smoothing or gradient scaling, with few achieving both through self-adaptive mechanisms.

VarCon's adaptive temperature mechanism provides bidirectional adaptation through the interaction between KL divergence and confidence-based temperature scaling. For samples with low confidence, the mechanism produces sharper target distributions (lower $\tau_2$) and stronger gradients, while confident samples receive smoother targets (higher $\tau_2$) and gentler gradients. This emerges naturally from our variational formulation without requiring manual scheduling or validation set monitoring.

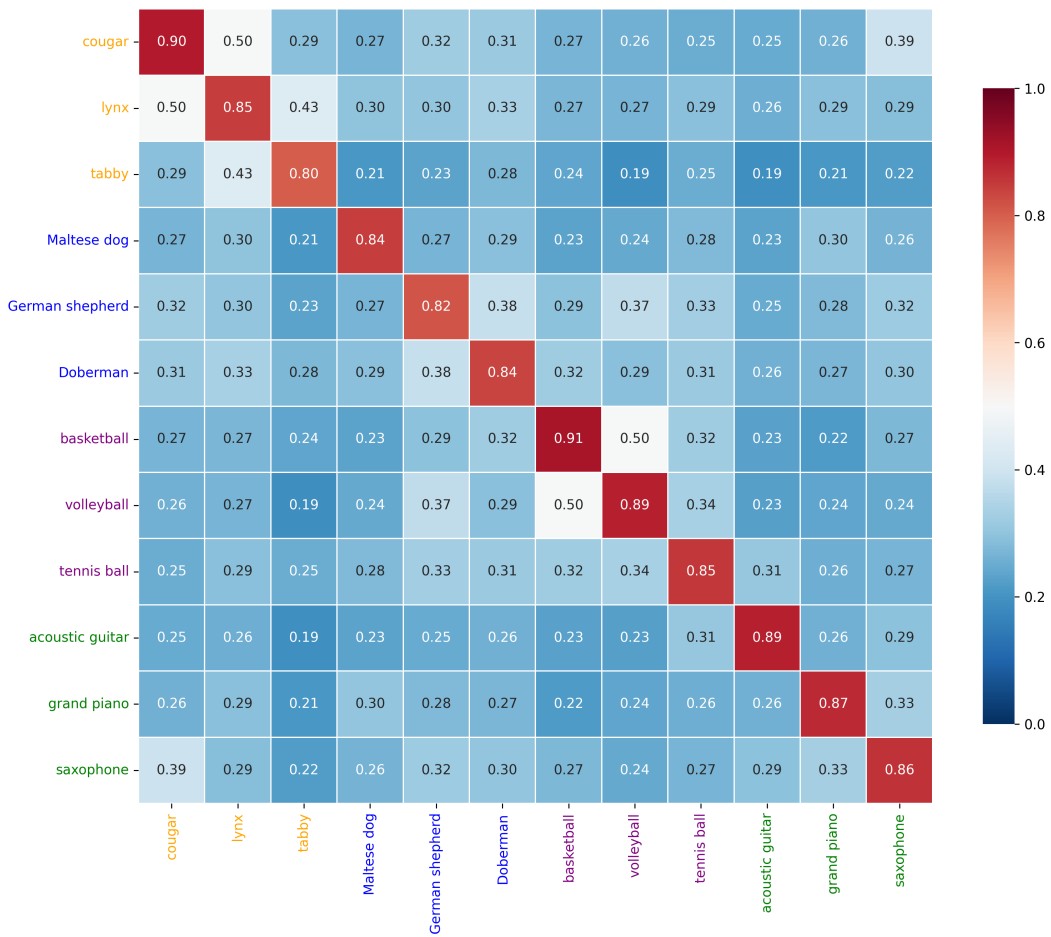

Figure 9: Learned cosine similarity matrix between class embeddings on ImageNet using VarCon with ResNet-50. The matrix visualizes pairwise similarities among 12 representative classes from four semantic groups: felines (cougar, lynx, tabby), dogs (Maltese, German shepherd, Doberman), balls (basketball, volleyball, tennis ball), and musical instruments (acoustic guitar, grand piano, saxophone). Higher values indicate stronger learned semantic relationships between classes based solely on visual features.

The scarcity of bidirectional self-adaptive methods in the literature reflects inherent optimization challenges: simultaneous target and gradient modifications can create conflicting optimization objectives or redundant regularization effects. Our approach addresses these challenges through the principled integration of temperature adaptation within the ELBO framework, where both modifications arise from a unified objective rather than separate mechanisms. This integration eliminates the need for balancing multiple hyperparameters while maintaining theoretical consistency with the variational objective.

# E    Additional Related Work

## E.1    Multi-Modal and Single-Modal Approaches

X-CLR [55] represents a recent advancement that leverages multi-modal information for contrastive learning. Table 10 summarizes the key methodological differences between SupCon, X-CLR, and VarCon.

These methodological differences highlight distinct design philosophies. SupCon extends self-supervised contrastive learning to the supervised setting by treating all same-class samples as positive

Table 10: Methodological comparison of supervised contrastive learning approaches.

| Aspect | SupCon | X-CLR | VarCon |
|---|---|---|---|
| Data Requirements | Single modality (images) | Dual modality (images + text) | Single modality (images) |
| Label Usage | Class labels for defining positive/negative pairs | Class names converted to text descriptions for similarity | Class labels for computing centroids and defining target distributions |
| Similarity Construction | Binary (same class = 1, different class = 0) | Pre-computed similarity matrix from text encoder cosine similarities | Dynamic distribution $q_\phi(r'|z)$ adjusted by confidence $p_\theta(r|z)$ |
| Theoretical Foundation | Heuristic extension of self-supervised learning | Heuristic soft labeling replacing binary similarities | Variational inference with ELBO maximization over latent variables |
| Objective Function | InfoNCE with multiple positives per class | Modified InfoNCE with soft similarity targets from text | KL divergence + log-posterior terms derived from ELBO |
| Computational Requirements | Single encoder + pairwise comparisons | Dual encoder (vision + text) + similarity matrix computation | Single encoder + class centroid computation |
| Applicability | Any labeled visual dataset | Requires meaningful text descriptions for classes | Any labeled visual dataset |

Table 11: Core innovations and target settings of contrastive methods.

| Method | Core Innovation | Target Setting |
|---|---|---|
| SupCon | Supervised contrastive loss | Balanced datasets |
| VarCon | Variational inference, ELBO objective | Balanced datasets |
| PaCo | Parametric centers + momentum encoder | Long-tailed datasets |
| GPaCo | Simplified parametric centers | Balanced and long-tailed datasets |

pairs, maintaining binary similarity relationships. X-CLR introduces graded similarities by leveraging semantic relationships encoded in class names through text encoders, enabling the model to learn that certain classes are more similar than others. VarCon takes a fundamentally different approach by reformulating the problem through variational inference, where the similarity construction emerges dynamically from the model's evolving confidence rather than being predetermined. This allows VarCon to adapt its supervision intensity per sample throughout training, providing stronger gradients for uncertain samples while relaxing constraints on well-learned examples. Notably, as shown in Figure 9, VarCon successfully learns semantically meaningful inter-class relationships purely from visual features, with related categories (e.g., different feline species or musical instruments) exhibiting higher similarities. While X-CLR requires auxiliary text modality to extract such semantic relationships, VarCon achieves adaptive learning using only visual data, making it applicable to any labeled dataset without requiring meaningful text descriptions. Although our variational framework could naturally extend to multi-modal settings, we first establish its effectiveness in single-modality scenarios to validate the theoretical contributions before exploring cross-modal applications.

### E.2 Synergy with Class-Imbalanced Methods

VarCon is designed as a general framework for supervised contrastive learning, focusing on improving representation quality through variational inference rather than addressing specific data distribution challenges. However, its theoretical foundation enables effective combination with specialized methods designed for class imbalance. PaCo [16] introduces parametric class centers maintained through exponential moving averages, specifically designed to handle long-tailed distributions where some classes have significantly fewer samples. GPaCo [17] simplifies this approach by removing the momentum encoder requirement while maintaining effectiveness on both balanced and imbalanced datasets. Table 11 illustrates how these methods address different aspects of the learning problem.

While VarCon focuses on general representation learning improvements, PaCo and GPaCo provide complementary mechanisms for handling class imbalance. Tables 12 and 13 demonstrate that when combined, VarCon's improved representation quality and PaCo/GPaCo's rebalancing strategies achieve superior results, suggesting that variational objectives and imbalance-handling are orthogonal dimensions of improvement. The performance gains from combining VarCon with PaCo/GPaCo validate that our variational framework provides a strong foundation that can be further enhanced with specialized techniques when dealing with imbalanced data distributions.

Table 12: Top-1 accuracy on ImageNet with ResNet-50.

| Method | Augmentation | Top-1 Acc |
| --- | --- | --- |
| SupCon | SimAugment | 77.82 |
| VarCon | SimAugment | 78.23 |
| PaCo | SimAugment | 78.70 |
| PaCo | RandAugment | 79.30 |
| GPaCo | RandAugment | 79.50 |
| VarCon+PaCo | SimAugment | 79.19 |
| VarCon+PaCo | RandAugment | 79.86 |
| **VarCon+GPaCo** | RandAugment | **79.94** |

Table 13: Top-1 accuracy on CIFAR-100 with ResNet-50.

| Method | Top-1 Acc |
| --- | --- |
| SupCon | 76.57 |
| VarCon | 78.29 |
| PaCo | 79.10 |
| GPaCo | 80.30 |
| VarCon+PaCo | 80.57 |
| **VarCon+GPaCo** | **81.29** |

