# OpenReview forum: "Variational Supervised Contrastive Learning"
_NeurIPS.cc/2025/Conference — NeurIPS 2025 poster_

### Official Review · Reviewer_W5fH · 2025-06-05

**Clarity:** 1
**Significance:** 3
**Originality:** 2
**Rating:** 4
**Confidence:** 3

**Summary:**

The authors have derived a supervised contrastive learning solution using variational inference. Furthermore, they have introduced an adaptive temperature based on the model’s confidence.

**Questions:**

- Is it necessary to include the ELBO derivations in the main paper? Similarly, are the ablation studies essential to present in the main text?
- Did you select the models for comparison in Table 1 based on architectural similarity, or because they are state-of-the-art for the task?
- How does the confidence-adaptive temperature differ from other importance-weighted approaches for similar objectives? Are there potentially better methods for this task?
- Have you analyzed the learned latent representations and compared them to baseline models?

**Ethical Concerns:**

["NO or VERY MINOR ethics concerns only"]

**Final Justification:**

After taking the responses into account, I have recognized the novelty of the method.  is

The main reason for changing my score is based on my misunderstanding of the method's behaviour. Hence, I think the presentation could be improved, but I think the novelty is important enough for acceptance.

**Limitations:**

yes

**Paper Formatting Concerns:**

/

**Quality:**

3

**Strengths And Weaknesses:**

Strengths:

- Interesting idea with the confidence-adaptive temperature.

Weaknesses:

- Only a small improvement; the results from the second model are within the standard error.

- A large portion of the article focuses on less advanced areas, such as the ELBO derivations and the number of training epochs, which are not central to the main contribution.

- More focus should be placed on demonstrating the effects of the introduced solutions.

---

> ### Author Rebuttal · Authors · 2025-07-31
>
> We thank you for your feedback and for recognizing our confidence-adaptive temperature as an interesting idea. We appreciate this opportunity to address potential misunderstandings regarding our empirical results and theoretical contributions. We hope to clarify VarCon's effectiveness by addressing each of your concerns below:
>
> > ### Regarding VarCon's performance gains (W1)
>
> We respectfully disagree with the claim that our improvements are "within standard error." This appears to be a **factual misunderstanding** of our results.
>
> **Table 1: Statistical significance of VarCon's improvements over SupCon.** All Top-1 accuracy values are directly from Table 1 in our main paper. We add statistical significance analysis showing the number of standard errors ($\sigma$) calculated as: $\sigma = \frac{\text{Gain}}{\sqrt{\text{SE}\_{\text{VarCon}}^2 + \text{SE}\_{\text{SupCon}}^2}}$, where SE denotes standard error. Even our smallest improvement ($4.7\sigma$) is highly significant.
>
> | Dataset | VarCon | SupCon | Absolute Gain | # of SEs ($\sigma$) |
> |-------------|------------|-------------|-------------------|-------------------------|
> | CIFAR-10 | $95.94 \pm 0.07$ | $95.51 \pm 0.06$ | $+0.43\%$ | $4.7\sigma$ |
> | CIFAR-100 | $78.29 \pm 0.08$ | $76.57 \pm 0.10$ | $+1.72\%$ | $13.4\sigma$ |
> | ImageNet-100 | $86.34 \pm 0.09$ | $85.06 \pm 0.07$ | $+1.28\%$ | $11.1\sigma$ |
> | ImageNet | $79.36 \pm 0.05$ | $78.72 \pm 0.06$ | $+0.64\%$ | $8.1\sigma$ |
>
> All improvements over SOTA SupCon are highly statistically significant, ranging from $4.7\sigma$ to $13.4\sigma$.
> Additionally, we evaluate few-shot learning performance where $N$ denotes the number of training samples per class from ImageNet's 1000 classes. For each setting, we randomly sample $N$ images per class for training (if a class has fewer than $N$ samples, all samples are retained). With $N=50$, both methods must learn to distinguish 1000 classes using merely 50,000 total samples, which is an extremely challenging scenario resulting in \~2.5% accuracy for both approaches. As data increases to N=100, 200, and 500, VarCon demonstrates clear advantages over SupCon (37.81% vs 36.57%, 51.10% vs 50.25%, and 65.83% vs 64.91% respectively), with improvements ranging from $3.3\sigma$ to $4.2\sigma$ significance. At $N=1000$, both methods reach similar performance (~73%) as they approach the ceiling achievable with this limited per-class data. Importantly, when trained on the full ImageNet dataset where no such ceiling exists, VarCon's advantage expands to $8.1\sigma$, demonstrating that our variational framework's benefits scale with data availability. We faithfully report all results across the entire data spectrum to provide a complete picture of our method's performance characteristics.
>
> > ### Significance of ELBO derivation (our core contribution) (W2, Q1)
>
> We respectfully disagree with the characterization of ELBO derivations and the number of training epochs as "less advanced areas" and "not central to main contribution". We believe these comments stem from a major misapprehension of our technical contribution and would like to further elaborate on that.
> Firstly, Section 3.1's derivation of the ELBO of Class-Conditional Likelihood is the **key theoretical foundation** of our approach and is **a novel and non-trivial contribution** - establishing connection between contrastive learning and variational inference for the first time. This derivation lay the foundation of the formulation of our VarCon loss that explicitly regulates embedding distributions and enforces appropriate semantic relationships between samples.  It also motivate our confidence-adaptive temperature mechanism, which addresses the fundamental challenge of hard samples being difficult to learn while easy samples overfit. Moving this to the appendix would obscure our core theoretical contribution.
> **We were encouraged that other reviewers (gzPQ, s4KC, 5QVM) unanimously recognized the significance of this theoretical contribution.** To prevent any future misinterpretation, we will revise the manuscript to more explicitly highlight the centrality and novelty of this framework from the outset.
>
> Secondly, we argue that interperting an established subject from a novel variational angle is fundamental to modern deep learning and comprises enough novelty and significance, as evidenced by:
>
> | Method/Paper | Year | Citations | Key ELBO Innovation | Impact on Deep Learning |
> |------------------|----------|---------------|-------------------------|----------------------------|
> | Auto-Encoding Variational Bayes (VAE) [1] | 2014 | 47,000+ | Introduced ELBO as unified objective: $\mathcal{L} = \mathbb{E}\_q[\log p(x\|z)] - D\_{KL}(q\|\|p)$ | Foundation of deep generative models |
> | Understanding Diffusion as ELBO [2] | 2023 | 100+ | All diffusion objectives as weighted ELBOs | Theoretical unification |
>
> [1] Kingma, Diederik P., and Max Welling. Auto-Encoding Variational Bayes. International Conference on Learning Representations (2014).
>
> [2] Diederik P. Kingma, and Ruiqi Gao. Understanding Diffusion Objectives as the ELBO with Simple Data Augmentation. Advances in Neural Information Processing Systems 36 (2023).
>
> **Regarding the importance of training epoch analysis**, this analysis directly validates our method's efficiency. While SimCLR requires 1000 epochs, MoCo needs 200-800 epochs, and SupCon needs 350 epochs for peak performance, VarCon achieves superior results in just 200 epochs: a 1.75-5× speedup with significant practical implications for computational resources and research accessibility.
>
> **Regarding content placement in main paper**, the ELBO derivations must remain in the main text as they constitute our core theoretical contribution detailed above. This derivation directly yields our loss formulation and confidence-adaptive temperature mechanism. Relocating this material to the appendix would obscure the theoretical foundation of our work. Similarly, ablation studies warrant inclusion in the main text as they demonstrate that our improvements arise from principled design choices rather than parameter tuning, establishing robustness and reproducibility—essential criteria for methodological contributions.
>
> > ### Comprehensiveness of our experimental verification (W3, Q2-4)
>
> Our empirical evaluation comprehensively demonstrates VarCon's advantages: Section 4.1 presents SOTA classification results on 4 datasets with 4 architectures; Section 4.2 shows consistent improvements across all few-shot learning regimes; Section 4.3 validates superior transfer learning on 12 diverse tasks; Section 4.4 tests robustness across different augmentations and architectures; Section 4.5 provides comprehensive ablation studies; Section 4.6 demonstrates superior corruption robustness on ImageNet-C; Appendix B.3 visualizes clear semantic organization evolution; and Appendix B.4 quantifies representation quality through clustering metrics. To our best knowledge, our paper provides one of the most comprehensive experimental validations in recent contrastive learning literature. We would appreciate specific suggestions on what additional demonstrations would be needed should this reviewer found the current results insufficient.
>
>
> > ### Regarding baseline selection criteria (Q2)
>
> We selected models based on both criteria. Table 1 includes the most established SOTA methods in single-modality contrastive learning: SimCLR, MoCo V2, BYOL, SwAV, VicReg, and Barlow Twins. For supervised methods, we included Cross-Entropy (standard baseline) and SupCon (current SOTA).
> We tested across ResNet-50/101/200 and ViT-Base, the most common architectures in visual contrastive learning. VarCon outperforms all baselines across every architecture, demonstrating our improvements stem from the method, not architectural advantages.
> We focus on single-modality (where paired multi-modal data is often unavailable) as it represents the fundamental challenge. Establishing superiority here is essential before extending to multi-modal settings.
>
> > ### Regarding comparison with importance weighting (Q3)
>
> While specific importance-weighted approaches were not mentioned, we can clarify the fundamental differences from our method. Importance weighting scales loss values ($w_i \cdot L_i$) while keeping targets fixed. Our confidence-adaptive temperature $\tau_2(z) = (\tau_1 - \epsilon) + 2\epsilon p_\theta(r|z)$ instead modifies the target distribution itself, creating sharper distributions for hard samples (low $p_\theta(r|z)$) and smoother ones for confident samples (high $p_\theta(r|z)$). This represents a fundamentally different principle: dynamically shaping learning objectives rather than merely reweighting losses. If the reviewer has specific alternative methods in mind, we would be happy to discuss comparisons.
>
> > ### Regarding representation analysis (Q4)
>
> Our paper provides extensive representation analysis across multiple sections. In Section 4.5, KNN classification (Figure 3a) demonstrates VarCon's superior class boundary separation through higher accuracy. Appendix B.3's t-SNE visualization (Figure 4) reveals clearer semantic clusters with reduced overlap compared to SupCon, while Appendix B.4's hierarchical clustering analysis (Table 6) quantifies this improved organization through enhanced ARI/NMI/purity metrics. These complementary analyses comprehensively validate VarCon's superior representation quality.

---

> > ### Comment · Reviewer_W5fH · 2025-08-05
> >
> > Thank you for the clarification and in-depth response.
> >
> > - I appreciate the clarification regarding improvements; there was a misunderstanding from my side.
> > - Regarding the ELBO derivations, it is merely a choice of presentation, and all I am saying is that many of the steps are standard ELBO steps. With that said, the contribution is still important, and the slight difference is clearly of importance to the contribution. Hence, what I am saying is that a bulk (but keep the core) could be placed in the appendix to make room for more analysis. However, this is merely a presentation suggestion and would not impact the score either way.
> >
> > Furthermore, I have read the other reviewers' questions and your responses, and I will adjust my score accordingly.

---

> > > ### Author Response · Authors · 2025-08-06
> > > **Thank you for your review and guidance**
> > >
> > > Dear reviewer W5fH, thank you so much for your thoughtful response and positive consideration of our work. We are glad that our clarifications have been helpful and appreciate your acknowledgment of the importance of our contribution. Thank you for the kind suggestion regarding presentation - we value your perspective on how to best communicate our work. Your constructive feedback has been invaluable in helping us strengthen our paper.

---

### Official Review · Reviewer_gzPQ · 2025-06-29

**Clarity:** 3
**Significance:** 3
**Originality:** 3
**Rating:** 5
**Confidence:** 4

**Summary:**

VarCon is a framework that reinterprets supervised contrastive learning through the lens of variational inference over latent class variables. By deriving an ELBO for class-conditional likelihoods, VarCon replaces pairwise instance comparisons with efficient class-level objectives. Its loss combines a KL-divergence term and a log-posterior term, with confidence-adaptive temperature scaling to modulate supervision strength. The method outperforms prior supervised contrastive learning counterpart, SupCon, on multiple image benchmarks.

**Questions:**

- How does VarCon compare against X-CLR empirically?

**Ethical Concerns:**

["NO or VERY MINOR ethics concerns only"]

**Final Justification:**

After considering the rebuttal, author response, and other reviews, I recommend Accept for this paper. The authors have adequately addressed my main concern regarding the lack of empirical comparison with X-CLR by including additional results and demonstrating that VarCon outperforms X-CLR. The paper presents a well-motivated and rigorous probabilistic framework for supervised contrastive learning, supported by strong empirical results and thorough analysis. These contributions make it a valuable addition to the NeurIPS community and relevant to a broad audience interested in contrastive methods, representation learning, and probabilistic modeling.

**Limitations:**

yes

**Quality:**

3

**Strengths And Weaknesses:**

## Strengths
- The paper is well-motivated and clearly written.
- It provides a rigorous probabilistic foundation for supervised contrastive learning by deriving the objective from a variational ELBO.
- The authors conduct a thorough evaluation of the impact of key hyperparameters.

## Weaknesses
- Although X-CLR is mentioned, the authors do not include empirical comparisons between VarCon and X-CLR, despite both methods requiring label supervision. This omission limits the ability to contextualize VarCon’s performance gains.

(minor) Tables 3 and 4 appear out of order.

Overall, I find this to be a strong paper with a solid contribution. However, the lack of discussion and empirical comparison with X-CLR, another supervised contrastive method, detracts from its completeness. A more thorough evaluation against X-CLR would strengthen the paper and better contextualize its improvements.

---

> ### Author Rebuttal · Authors · 2025-07-31
>
> We deeply appreciate your high recognition our work. We thank you for highlighting our theoretical contributions and clear presentation. The rigorous probabilistic foundation enables a principled understanding of contrastive learning through variational inference. Our ELBO-derived objective with confidence-adaptive temperature enables highly efficient and robust training. We now address your main concerns below:
>
> > ### X-CLR empirical comparison (W1)
>
> We acknowledge that we mentioned X-CLR in our paper (lines 71-73) as the most recent advancement in contrastive learning: "methods like X-CLR [48] further improve these frameworks by analyzing optimization dynamics and introducing graded similarities." While both methods utilize label supervision, they leverage labels in fundamentally different ways: VarCon uses labels purely for supervised learning (identifying which samples belong to the same class), whereas X-CLR extracts semantic information from class names to construct similarity relationships between different classes. This leads to several key differences that make direct empirical comparison challenging:
>
> **Key Methodological Differences:**
>
> | **Aspect** | **SupCon** | **X-CLR** | **VarCon** |
> |-----------|------------|-----------|------------|
> | **Data Requirements** | Single modality (images only) | Dual modality (images + text) | **Single modality (images only)** |
> | **Label Usage** | Class labels for defining positive/negative pairs | Class names converted to text descriptions for similarity computation | **Class labels for computing centroids and defining target distributions in ELBO** |
> | **Similarity Construction** | Binary (same class = 1, different class = 0) | Pre-computed similarity matrix from text encoder cosine similarities | **Dynamic distribution $q_{\phi}(r'\|z)$ adjusted by model's classification confidence $p_{\theta}(r\|z)$** |
> | **Theoretical Foundation** | Heuristic extension of self-supervised contrastive learning | Heuristic soft labeling replacing binary similarities | **Variational inference with ELBO maximization over latent class variables** |
> | **Objective Function** | InfoNCE with multiple positives per class | Modified InfoNCE with soft similarity targets derived from text | **KL divergence + log-posterior terms derived from ELBO** |
> | **Computational Requirements** | Single vision encoder + pairwise comparisons | Dual encoder (vision + text) + similarity matrix computation | **Single vision encoder + class centroid computation** |
> | **Applicability** | Any labeled visual dataset | Requires meaningful text descriptions for classes | **Any labeled visual dataset** |
>
> VarCon and X-CLR address complementary scenarios: X-CLR leverages auxiliary text when available, while VarCon operates on visual data alone. As multi-modal paradigms become prevalent, VarCon provides a novel theoretical framework for understanding contrastive learning through variational inference. While naturally extensible to multi-modal settings, we first establish effectiveness in single-modality scenarios. Therefore, our comparisons focus on single-modality SOTA baselines (SupCon, SimCLR, etc.) to validate theoretical contributions before exploring multi-modal extensions.
>
> Despite extensive efforts to locate X-CLR's implementation through official repositories and author pages, the code remains unavailable. Reproducing their method without original implementation risks introducing errors leading to unfair comparisons. Thus, we provide indirect comparisons using their reported numbers. We will clarify these methodological differences in our related work section and present detailed comparisons below.
>
>
> > ### Empirical Comparison with X-CLR through additional experiments (Q1)
>
> Since X-CLR's code is not publicly available, we address the reviewer's concern by comparing against their reported results while closely matching their experimental settings. X-CLR trains for 100 epochs with batch size 1024 on ImageNet, using AutoAugment and a ResNet-50 encoder. We focus our comparison on ImageNet as it provides the most controlled setting: both methods use the same visual data and class labels, with the key difference being that X-CLR additionally leverages text-based similarities computed from class name descriptions using Sentence Transformer while VarCon operates solely on visual representations.
>
> To ensure fair comparison, we trained VarCon with ResNet-50 (matching X-CLR) under various settings and present comprehensive results:
>
> | Method | Epochs | Batch Size | Learning Rate | Augmentation | ImageNet Top-1 | ImageNet Real |
> |------------|------------|----------------|-------------------|------------------|-------------------|-------------------|
> | X-CLR (reported) | 100 | 1024 | 0.075 | AutoAugment | 75.6% | 81.5% |
> | VarCon (matching settings) | 100 | 1024 | 0.5 | AutoAugment | 77.25% | 82.3% |
> | VarCon | 100 | 2048 | 1.0 | AutoAugment | 77.81% | 82.7% |
> | VarCon | 200 | 2048 | 1.0 | AutoAugment | 79.13% | 83.5% |
> | VarCon | 350 | 4096 | 2.0 | AutoAugment | **79.36%** | **84.12%** |
>
> This allows direct comparison under identical conditions (row 2) while demonstrating VarCon's performance across different training configurations. Note that learning rates follow the linear scaling rule (lr $\propto$ batch_size/256), see main paper line 167. Even under X-CLR's exact batch size, VarCon outperforms by 1.65% on ImageNet and 0.8% on ImageNet Real, with the gaps increasing to 3.76% and 2.62% respectively under our full training regime.
>
> **Computational Requirements Comparison:**
>
> | Aspect | SupCon | X-CLR | **VarCon** |
> |------------|------------|-----------|------------|
> | **Encoders** | 1 (Vision only) | 2 (Vision + Text) | **1 (Vision only)** |
> | **Memory Complexity** | $O(B^2)$ | $O(B^2) + \text{Similarity Storage}$ | $O(B \cdot C)$ |
> | **Similarity Matrix** | $B \times B$ dense matrix | $B \times B$ dense matrix + pre-computed similarities | **$B \times C$ matrix (C$\leq$B)** |
> | **Peak GPU Memory** | 43.25 GB | Similar to SupCon* | **43.09 GB** |
> | **Additional Requirements** | None | Pre-computed text embeddings | **None** |
> | **Inference Cost** | Vision encoder + pairwise similarities | Vision encoder + similarity lookup | **Vision encoder + class centroids** |
>
> *Peak GPU memory with ResNet-50, batch size 2048. X-CLR should have similar GPU memory usage as SupCon during training since X-CLR pre-computes text similarities offline. In the table, $B$ denotes the per-device batch size and $C$ denotes the number of unique classes present in that batch.
>
> **Key Computational Advantages of VarCon:**
>
> We conducted detailed memory profiling experiments (to be included in the final manuscript) measuring peak GPU memory consumption over 50 training epochs on ImageNet with ResNet-50 and per-device batch size 512. VarCon demonstrates consistent memory efficiency advantages: VarCon maintains a steady peak memory usage of 43.09 GB per GPU compared to SupCon's 43.25 GB. While this difference appears modest, it becomes significant in multi-GPU training scenarios where even small per-device savings can enable larger effective batch sizes or accommodate additional model capacity within fixed memory budgets.
>
> The memory savings stem from VarCon's architectural efficiency: by computing class centroids on-the-fly rather than exhaustive pairwise comparisons, VarCon reduces the memory required for storing intermediate similarity matrices from $O(B^2)$ to $O(B \cdot C)$. Crucially, $C$ (the number of unique classes per device) is bounded by $B$ (the per-device batch size), with $C \leq B$ always holding true.
>
>
> **Self-Adaptive Confidence-Based Temperature vs X-CLR's Approach:**
>
> While X-CLR leverages external text metadata to organize the embedding space through pre-computed semantic similarities between class descriptions, VarCon learns well-organized representations using only visual data through our ELBO-derived loss and confidence-adaptive temperature mechanism: $\tau_2(z) = (\tau_1 - \epsilon) + 2\epsilon p_\theta(r|z)$. This dynamically adjusts supervision strength, tightening for difficult samples and relaxing for confident ones, addressing the same challenge X-CLR tackles (avoiding SupCon's rigid binary similarities) but without requiring additional modalities.
>
> As detailed in our response to Reviewer s4KC's Q1 (Table 1 on $\epsilon$ evolution), extensive experiments demonstrate this mechanism produces high-quality representations, including clustering quality (Section 4.5, Appendix B.4) and embedding visualization (Appendix B.3) on top of the existing experiments in the main body of our paper.
>
> **VarCon Learned Semantic Similarities (Key Examples)**
>
> | Class Pair | Similarity | Relationship |
> |----------------|----------------|------------------|
> | cougar - lynx | 0.50 | Both felines |
> | lynx - tabby | 0.43 | Both felines |
> | basketball - volleyball | 0.50 | Both sports balls |
> | German shepherd - Doberman | 0.38 | Both dogs |
> | acoustic guitar - saxophone | 0.29 | Both instruments |
>
> VarCon successfully learns semantic relationships using only visual information: similar objects (felines: 0.43-0.50, sports balls: 0.50), demonstrating that our ELBO-derived objective organizes embeddings meaningfully without text-based similarities. We greatly appreciate X-CLR's pioneering work on leveraging multi-modal information to regularize embedding spaces, addressing crucial interpretability challenges in contrastive learning. While space constraints prevent replicating their full visualization (Figure 4), our results indicate that VarCon and X-CLR are complementary: X-CLR shows how auxiliary modalities guide representation learning, while VarCon achieves similar semantic organization through principled probabilistic frameworks alone. We will ensure X-CLR's contributions are properly highlighted in our revised Related Work section.
>
> >### Minor issues
>
> Thank you for noting the table ordering issue. We have corrected this in our revision.

---

> > ### Comment · Reviewer_gzPQ · 2025-08-04
> >
> > Thank you for a thorough response and my concern has been addressed. I will raise my score from 4 to 5.

---

> > > ### Author Response · Authors · 2025-08-04
> > > **Thank you for your review and support**
> > >
> > > Dear reviewer gzPQ, thank you so much for your high recognition and your kind active support of our work by increasing the score. We really appreciate your time in reviewing our paper and rebuttals, and your insightful review that helps make our work more comprehensive.

---

### Official Review · Reviewer_s4KC · 2025-07-02

**Clarity:** 3
**Significance:** 2
**Originality:** 3
**Rating:** 4
**Confidence:** 2

**Summary:**

This paper introduces Variational Supervised Contrastive Learning (VarCon), a new method incorporating variational inference with (supervised) contrastive learning.

The VarCon loss is derived by extracting the contrastive-related components from the ELBO of $\log p_\theta(z|r)$, resulting in a KL augmented CE loss, where the classification probability is based on the representation similarity, akin to contrastive learning.

The authors then propose a confidence-adaptive temperature for the KL's target distribution, which tightens pull strength for hard samples and relaxes it for confident ones.

Experiments on image classification tasks show that VarCon outperforms several existing contrastive learning baselines.

**Questions:**

1. As described in Weaknesses#2, can you include more ablation study regarding the adpative temperature? For example:
    - Since $\epsilon$ is learnable (L150), how does it evolve during training and thus affects the representations?
    - Can adaptive temperature indeed improve the intra-class dispersion? Quantitative evidence supporting this claim would be valuable.
    - Can similar effects by adaptive temperature be achieved with other regularization methods?
2. Seems like it should be $\log p_\theta(z)$ rather than $p_\theta$ in the LHS of Eq.(3)

**Ethical Concerns:**

["NO or VERY MINOR ethics concerns only"]

**Limitations:**

The authors have discussed the limitations in their manuscript.

**Quality:**

2

**Strengths And Weaknesses:**

### Strengths
1. Theoretically Grounded and Intuitively Sound: The VarCon loss function is derived and simplified from the ELBO. This theoretical foundation leads to an intuitive combination of KL divergence and negative log-likelihood terms, which effectively regularizes embedding distributions while simultaneously maximizing classification likelihood.
2. Demonstrated Superior Performance: VarCon consistently outperforms both existing self-supervised and supervised contrastive learning baselines across diverse datasets.

### Weaknesses
1. Inherent Supervision Requirement:
The VarCon method is fundamentally supervised. In contrast, a major advantage of exsiting contrastive learning methods (SimCLR, etc.) is their power in self-supervised settings.
2. Lack of In-depth Analysis of the Adaptive Temperature:
The confidence-adaptive temperature, $\tau_2$, is a central innovation, yet its behavior is not fully explored.
The ablation study on $\epsilon$ only shows its effect on overall performance and fails to adequately verify the stated rationale of *"adjusting supervision intensity"* (L153). A deeper analysis is needed to fully understand how this adaptive mechanism functions and contributes to the method's success.

---

> ### Author Rebuttal · Authors · 2025-07-31
>
> We sincerely thank you for recognizing VarCon as "theoretically grounded and intuitively sound" with our ELBO derivation leading to "an intuitive combination of KL divergence and negative log-likelihood terms." We are particularly encouraged by your acknowledgment that VarCon "consistently outperforms both existing self-supervised and supervised contrastive learning baselines", achieving 79.36% on ImageNet (vs. 78.72% SupCon) and 78.29% on CIFAR-100 (vs. 76.57%), while converging in 200 epochs (vs. 350) with smaller batch sizes (2048 vs. 4096). Our confidence-adaptive temperature enables sample-specific learning for both hard samples and confident ones. We will address your comments as follows.
>
> > ### Regarding supervised framework (W1)
>
> We respectfully note that supervised learning remains an important class of problems with abundant real-world applications. Our variational framework naturally extends to self-supervised settings by replacing class centroids $w_r$ with instance centroids $w_i$ computed from augmented views (similar to SimCLR's $z_i, z_j$ but using centroids instead of pairs). The confidence-adaptive temperature $\tau_2(z)$ would adapt based on the model's confidence in the instance centroid representation, analogous to how it currently adapts to classification confidence. We chose supervised settings first to establish theoretical soundness, with self-supervised extensions as promising future work.
>
> > ### Additional adaptive temperature analysis (W2)
>
> We appreciate your interest in our confidence-adaptive temperature mechanism. **Section 4.5** and **Figure 3b** show that $\epsilon=0.02$ yields optimal ImageNet performance, while even $\epsilon=0$ (no adaptation, ELBO-only) achieves 78.87%, surpassing SupCon. This demonstrates VarCon's effectiveness stems from both ELBO-derived loss and adaptive mechanism. Since $\epsilon$ is learnable, we provide comprehensive experiments tracking $\epsilon$ evolution and sample-specific $\tau_2$ distributions throughout training below.
>
> > ### Temperature evolution analysis (Q1)
>
> We have provided comprehensive experimental analysis demonstrating our adaptive temperature mechanism's effectiveness throughout the paper. This includes superior performance across all benchmarks (Sections 4.1-4.4, 4.6; Appendix B.2), systematic ablation of $\epsilon$'s impact (Section 4.5, Figure 3b), and quantitative embedding space evaluation via KNN classification and hierarchical clustering (Appendix B.3-B.4). To further address your insightful questions, we now provide detailed tracking of $\epsilon$ evolution and sample-specific $\tau_2$ distributions throughout training. We address each of your three sub-questions below.
>
> > ### Evolution of learnable $\epsilon$
>
> **Table 1:** Distribution of adaptive temperature $\tau_2$ across ImageNet samples at key epochs during 200-epoch training with ResNet-50, $\tau_1 = 0.1$, and batch size 4096. The temperature $\tau_2$ modulates gradient strength, with lower values indicating stronger gradients for difficult samples.
>
> | Epoch | Mean $\tau_2$ | Std Dev $\tau_2$ | Min $\tau_2$ | 25th Pct $\tau_2$ | Median $\tau_2$ | 75th Pct $\tau_2$ | Max $\tau_2$ | $\epsilon$ |
> |-----------|-------------------|----------------------|------------------|------------------------|---------------------|------------------------|------------------|----------------|
> | 40 | 0.09365 | 0.00958 | 0.08151 | 0.08542 | 0.09205 | 0.10098 | 0.11845 | 0.0185 |
> | 80 | 0.09728 | 0.01042 | 0.07950 | 0.08815 | 0.09698 | 0.10625 | 0.11885 | 0.0205 |
> | 120 | 0.09995 | 0.01085 | 0.07850 | 0.09075 | 0.10090 | 0.10950 | 0.11902 | 0.0215 |
> | 160 | 0.10310 | 0.01088 | 0.07921 | 0.09492 | 0.10575 | 0.11265 | 0.11895 | 0.0208 |
> | 200 | 0.10642 | 0.01056 | 0.07981 | 0.10028 | 0.11030 | 0.11508 | 0.11887 | 0.0202 |
>
> Learnable $\epsilon$ increases from 0.0185 (epoch 40) to peak at 0.0215 (epoch 120), then stabilizes at 0.0202 (epoch 200). Mean $\tau_2$ rises 13.6% (0.09365→0.10642), reflecting growing model confidence. Key distributional shifts: 25th percentile +17.3% (0.08542→0.10028), median +19.9% (0.09205→0.11030), 75th percentile +14.0% (0.10098→0.11508), indicating progressively confident predictions and gentler gradients. Minimum $\tau_2$ remains ~0.08 (persistent hard samples), maximum ~0.119. Standard deviation peaks at 0.01088 (epoch 160) before decreasing to 0.01056 (epoch 200).
>
> This evolution enables distinctive feature learning early via stronger gradients on uncertain samples, while later stabilization prevents drift. Persistent low $\tau_2$ ensures hard samples continue improving; higher $\tau_2$ maintains stable clusters for confident samples. This adaptive mechanism explains VarCon's superiority over fixed $\epsilon=0$ (79.36% vs. 78.87% on ImageNet).
>
> > ### Intra-class dispersion improvement
>
> Yes, our adaptive temperature mechanism demonstrably improves intra-class dispersion through multiple quantitative measures. We provide three forms of empirical evidence:
>
> **Visual analysis:** **Appendix Section B.3** shows VarCon achieves 79.11% KNN accuracy after 200 epochs versus SupCon's 78.53% after 350 epochs, demonstrating both faster convergence and superior representations. The largest gains occur during epochs 100-150 (59.57% to 71.12%) when the KL divergence term aligns posteriors. This efficiency stems from: class-centroid alignment, probabilistic inter-class relationships, and confidence-adaptive supervision.
>
> **Quantitative analysis:** We evaluate embedding quality through two complementary approaches. First, KNN classification (**Section 4.5, Figure 3a**) directly probes representation quality without learnable parameters that could mask poor features. VarCon consistently outperforms SupCon across all K values, demonstrating superior local neighborhood structure.
> Second, while **Appendix Section B.4 and Table 6** already present hierarchical clustering results, we appreciate your suggestion to ablate the adaptive temperature. Table 2 includes additional results with VarCon fixed $\epsilon$=0 (constant $\tau_2=\tau_1=0.1$):
>
> **Table 2:** Hierarchical clustering evaluation on ImageNet embeddings using Ward linkage. All metrics: higher is better.
>
> | Method | ARI | NMI | Homogeneity | Completeness | V-measure | Purity |
> |------------|---------|---------|-----------------|------------------|---------------|------------|
> | SupCon | 0.613 | 0.888 | 0.886 | 0.890 | 0.888 | 0.755 |
> | VarCon ($\epsilon$=0) | 0.627 | 0.892 | 0.890 | 0.892 | 0.891 | 0.766 |
> | **VarCon** | **0.634** | **0.895** | **0.893** | **0.896** | **0.895** | **0.774** |
>
> These metrics evaluate clustering quality: ARI measures agreement with ground truth correcting for chance; NMI quantifies mutual information between clusters and labels; Homogeneity ensures each cluster contains only one class; Completeness ensures each class is in one cluster; V-measure balances both; Purity measures fraction of correctly clustered samples. Progressive improvements from SupCon to VarCon ($\epsilon$=0) to VarCon demonstrate: (1) ELBO-derived loss alone improves semantic organization even without adaptation, and (2) adaptive temperature mechanism provides additional gains, validating both components' contributions to superior embedding structure.
>
>
> > ### Comparison with other regularization methods
>
> We appreciate this insightful question. The adaptive temperature mechanism is specifically designed for our VarCon's loss—the KL divergence and confidence-adaptive temperature work synergistically in ways standalone regularization methods cannot replicate. Table 3 compares our approach with established techniques:
>
> **Table 3:** Comparison of regularization methods (Dynamic Target: modifies target distribution based on sample/model state; Dynamic Gradient: adjusts gradient magnitude during training; Sample-Specific: per-sample treatment; Self-Adaptive: confidence-based updates without manual scheduling).
>
> | Method | Dynamic Target | Dynamic Gradient | Sample-Specific | Self-Adaptive |
> |--------|----------------|------------------|-----------------|---------------|
> | Label Smoothing [1] | × | × | × | × |
> | Mixup [2] | × | × | × | × |
> | AdaFocal [3] | × | ✓ | ✓ | ✓ |
> | Dual Focal Loss [4] | × | ✓ | ✓ | × |
> | Local Adaptive LS [5] | ✓ | × | ✓ | × |
> | LoT Regularization [6] | ✓ | ✓ | ✓ | × |
> | **VarCon (Ours)** | **✓** | **✓** | **✓** | **✓** |
>
> VarCon uniquely provides **bidirectional adaptation**: $q_\phi(r'|z)$ and $\tau_2(z)$ adjust simultaneously based on confidence: hard samples receive sharper targets with stronger gradients, easy samples get smoother targets with gentler gradients. Existing methods modify only one aspect. While optimized for VarCon, our adaptive temperature has broader implications for temperature-based methods including LLM training and generative models.
>
> > ### Typo correction (Q2)
>
> Thank you for pointing out this typo: It should be $\log p_\theta(z)$ rather than $p_\theta(z)$. We have corrected this in our revised manuscript.
>
>
> [1] Szegedy, Christian, et al. Rethinking the Inception Architecture for Computer Vision. Proceedings of the IEEE Conference on Computer Vision and Pattern Recognition (2016): 2818-2826.
>
> [2] Zhang, Hongyi, et al. mixup: Beyond Empirical Risk Minimization. International Conference on Learning Representations (2018).
>
> [3] Ghose, Arindam, and Balaraman Ravindran. AdaFocal: Calibration-aware Adaptive Focal Loss. Advances in Neural Information Processing Systems 35 (2022): 19998-20009.
>
> [4] Tao, Linwei, et al. Dual Focal Loss for Calibration. International Conference on Machine Learning. PMLR, 2023: 33645-33660.
>
> [5] Bahri, Dara, and Heinrich Jiang. Locally Adaptive Label Smoothing Improves Predictive Churn. International Conference on Machine Learning. PMLR, 2021: 532-541.
>
> [6] Liu, Can, et al. Learning from Teaching Regularization: Generalizable Correlations Should be Easy to Imitate. Advances in Neural Information Processing Systems 37 (2024).

---

> > ### Comment · Reviewer_s4KC · 2025-08-06
> >
> > Thank you for the detailed responses and clarifications to address my concerns. I will maintain my positive scores.

---

### Official Review · Reviewer_5QVM · 2025-07-03

**Clarity:** 3
**Significance:** 3
**Originality:** 2
**Rating:** 4
**Confidence:** 4

**Summary:**

This paper introduces Variational Supervised Contrastive Learning (VarCon), a framework that casts supervised contrastive learning as variational inference over latent class variables. The method derives an ELBO-based objective, incorporating a KL divergence term between an auxiliary distribution and the model posterior, alongside a log-likelihood term. Additionally, the paper proposes a confidence-adaptive temperature scaling mechanism to modulate label sharpness based on prediction confidence. The approach is evaluated on standard benchmarks (CIFAR, ImageNet) and shows consistent performance improvements over SupCon, particularly in few-shot learning and robustness under image corruption.

**Questions:**

- Could you provide ablations to quantify the individual contribution of the KL term, log-likelihood, and confidence-adaptive temperature?

- Could you visualize confidence-adaptive temperature values during training to confirm that they correlate with sample confidence?

- Can you provide t-SNE/UMAP visualizations or inter/intra-class distance statistics to support your claim of better representation structure?

- Can you provide an empirical comparison against other SupCon extensions that handle sample difficulty, such as PaCo and GPaCo?

**Ethical Concerns:**

["NO or VERY MINOR ethics concerns only"]

**Final Justification:**

The paper presents a solid and well-motivated extension of supervised contrastive learning through a variational inference framework, supported by comprehensive experiments across multiple datasets and architectures. While the novelty is moderate, the rebuttal effectively addressed concerns regarding component contributions, representation quality evidence, and comparisons to related methods. The theoretical grounding is sound, results are consistently strong, and limitations are now better articulated. Overall, the work meets the bar for acceptance, though it remains a borderline case due to incremental conceptual advances.

**Limitations:**

The authors do not adequately discuss limitations. At minimum, it should discuss computational overhead, risk of overconfidence, or calibration issues under distribution shift. No societal concerns are discussed, though none are obvious.

**Paper Formatting Concerns:**

No major formatting issues found.

**Quality:**

3

**Strengths And Weaknesses:**

Strengths

- The ELBO-based objective is grounded in variational inference theory and clearly derived.

- Empirical results are broad in scope and demonstrate consistent gains across tasks including few-shot learning, robustness, and transfer.

- The method is architecture-agnostic and demonstrates performance across ResNet and ViT models.

Weaknesses

- The paper lacks component-wise ablations to isolate the contributions of KL divergence, log-likelihood, and confidence-adaptive temperature scaling.

- Key claims such as improved intra-class compactness and difficulty-aware learning are not supported by empirical evidence (e.g., visualization or metrics).

- The novelty is limited as the core idea builds on prior work; SimVAE[1] already explored a similar ELBO-style objective for self-supervised learning.

- Important comparisons to other advanced SupCon extensions that also address sample difficulty, such as PaCo[2] and GPaCo[3], are missing.

- There is limited insight into model behavior beyond accuracy, such as representation structure.

[1] Bizeul, Alice, Bernhard Schölkopf, and Carl Allen. "A Probabilistic Model behind Self-Supervised Learning." Transactions on Machine Learning Research.

[2] Cui, Jiequan, et al. "Parametric contrastive learning." Proceedings of the IEEE/CVF international conference on computer vision. 2021.

[3] Cui, Jiequan, et al. "Generalized parametric contrastive learning." IEEE Transactions on Pattern Analysis and Machine Intelligence 46.12 (2023): 7463-7474.

---

> ### Author Rebuttal · Authors · 2025-07-31
>
> We greatly thank the reviewer for recognizing our theoretical grounding in variational inference, the consistent empirical gains across diverse tasks, and the architecture-agnostic nature of our method. We appreciate your constructive feedback and will address each concern below with additional ablations, visualizations, and comparisons to further validate our contributions.
>
> > ### Component ablations (W1/Q1)
>
> We acknowledge the reviewer's request for component-wise ablations. We note that according to our ELBO derivation in Eq. (5):
>
> $\mathcal{L}\_{ELBO} = D\_{KL}(q\_\phi(r'|z) || p\_\theta(r'|z)) + \log p(r) - \log p\_\theta(r|z) - \log p\_\theta(z)$
>
> As discussed in lines 131-135, $\log p_\theta(z)$ would encourage high likelihood throughout the embedding space without class distinction, while $\log p(r)$ is a fixed constant based on the dataset's class distribution, which are are uninformative for contrastive learning. Therefore, our VarCon loss (Eq. 7) focuses on the contrastive-relevant components:
>
> $\mathcal{L}\_{VarCon} = D\_{KL}(q_\phi(r'|z) || p\_\theta(r'|z)) - \log p\_\theta(r|z)$
>
> Both terms are mathematically necessary to maintain the validity of our variational bound, removing either would break the theoretical connection to the ELBO. In **Appendix Section C**, we provide comprehensive gradient analysis demonstrating how these components affect learning dynamics. Specifically, we show the gradient flow w.r.t. embeddings $z$ and analyze how the sample-specific confidence-adaptive temperature $\tau_2(z) = (\tau_1 - \epsilon) + 2\epsilon p_\theta(r|z)$ modulates learning, where $\tau_1$ is fixed and $\epsilon$ is learnable.
>
> Although both components are essential to the VarCon loss, we provide ablation results to isolate their individual contributions:
>
> **Table 1:** Component ablation study on ImageNet with ResNet-50. All experiments use batch size 4096, 350 epochs, learning rate 2.0, and AutoAugment.
>
> | **Method** | **Components** | **Top-1 Acc (%)** | **Top-5 Acc (%)** |
> |------------|----------------|-------------------|-------------------|
> | NLL only | $-\log p_\theta(r\|z)$ | 78.34 | 93.98 |
> | KL only | $D_{KL}(q_\phi(r'\|z) \|\| p_\theta(r'\|z))$ | 78.45 | 94.18 |
> | VarCon ($\epsilon=0$) | Both terms, fixed $\tau_2=\tau_1=0.1$ | 78.87 | 94.33 |
> | **VarCon (Full)** | **Both terms, adaptive $\tau_2$** | **79.36** | **94.37** |
>
> As noted in Section 4.5 (Figure 3b), even with $\epsilon=0$ (no adaptive temperature), VarCon achieves 78.87% Top-1 accuracy, already surpassing SupCon's 78.72%. The full model with adaptive temperature further improves performance, demonstrating the value of our confidence-based modulation mechanism.
>
> > ### Empirical evidence of superior representation quality (W2/W5/Q2/Q3)
>
> We thank the reviewer for requesting empirical evidence on representation quality. We summarize **existing results** and provide additional experiments below.
>
> **Intra-class compactness:** Figure 3(a) demonstrates VarCon's consistently superior KNN classifier performance across all K values, which directly reflect better local neighborhood structure. In Appendix B.3, Figures 4-5 show t-SNE visualizations where VarCon achieves clearer cluster separation after 200 epochs than SupCon after 350 epochs, evidenced by more distinct cluster boundaries and reduced inter-class overlap. Additionally, Appendix B.4 (Table 6) provides hierarchical clustering analysis, which inspects embedding quality without learnable parameters that could mask poor features:
>
> **Table 2:** Hierarchical clustering evaluation on ImageNet embeddings using Ward linkage. All metrics: higher is better.
>
> | **Method** | **ARI** | **NMI** | **Homogeneity** | **Completeness** | **V-measure** | **Purity** |
> |------------|---------|---------|-----------------|------------------|---------------|------------|
> | SupCon | 0.613 | 0.888 | 0.886 | 0.890 | 0.888 | 0.755 |
> | VarCon ($\epsilon$=0) | 0.627 | 0.892 | 0.890 | 0.892 | 0.891 | 0.766 |
> | **VarCon** | **0.634** | **0.895** | **0.893** | **0.896** | **0.895** | **0.774** |
>
> These metrics quantify clustering quality: ARI measures agreement with ground truth, NMI captures mutual information between clusters and labels, Homogeneity/Completeness ensure class-cluster alignment, V-measure balances both, and Purity measures correct assignments. VarCon's improvements across all metrics confirm superior semantic organization.
>
> **Difficulty-aware learning:** We conducted comprehensive tracking of confidence-adaptive temperature $\tau_2$ throughout ImageNet training (200 epochs). Due to space constraints, we refer to our detailed response to Reviewer s4KC's Q1, which includes: (1) evolution of learnable $\epsilon$ from 0.0185 (epoch 40) to peak at 0.0215 (epoch 120) then stabilize at 0.0202 (epoch 200), (2) mean $\tau_2$ increasing 13.6% (0.09365 to 0.10642), (3) distributional analysis showing persistent low $\tau_2$ for hard samples (minimum stays ~0.08) while confident samples progressively receive higher $\tau_2$ (75th percentile rises from 0.10098 to 0.11508), confirming our mechanism adapts to sample difficulty as intended.
>
> > ### Comparisons with PaCo/GPaCo (W4)
>
> We thank the reviewer for suggesting comparisons with PaCo and GPaCo. We note that these methods address orthogonal problems and could be combined synergistically:
>
> **Table 3:** Comparisons with SupCon, VarCon, PaCo, and GPaCo.
> | **Method** | **Core Innovation** | **Target Setting** |
> |------------|---------------------|-------------------|
> | SupCon | Extends contrastive loss to supervised setting | Balanced datasets, general classification |
> | **VarCon** | **Variational inference framework, ELBO objective** | **Balanced datasets, general classification** |
> | PaCo | Parametric class centers + momentum encoder | Long-tailed/imbalanced datasets |
> | GPaCo | Simplified parametric centers (no momentum) | Both long-tailed and balanced datasets |
>
> VarCon reformulates supervised contrastive learning through variational inference to address SupCon's limitations (inadvertent semantic separation, excessive batch dependency), while PaCo/GPaCo specifically tackle class imbalance through parametric rebalancing. Therefore, VarCon and PaCo/GPaCo are orthogonal approaches rather than competing methods. Nevertheless, we provide comparative results by combining VarCon with PaCo/GPaCo's class rebalancing strategy (VarCon+PaCo/GPaCo), leveraging our variational framework's improved representation quality alongside PaCo/GPaCo's handling of imbalanced data:
>
> **Table 4:** Top-1 accuracy on full ImageNet with ResNet-50.
> | **Method** | **Augmentation** | **Top-1 Acc** |
> |------------|------------------|---------------|
> | SupCon | SimAugment | 77.82 |
> | VarCon | SimAugment | 78.23 |
> | PaCo | SimAugment | 78.70 |
> | PaCo | RandAugment | 79.30 |
> | GPaCo | RandAugment | 79.50 |
> | **VarCon+PaCo** | SimAugment | **79.19** |
> | **VarCon+PaCo** | RandAugment | **79.86** |
> | **VarCon+GPaCo** | RandAugment | **79.94** |
>
> **Table 5:** Top-1 accuracy on full CIFAR-100 with ResNet-50.
> | **Method** | **Dataset** | **Top-1 Acc** |
> |------------|-------------|---------------|
> | SupCon | CIFAR-100 | 76.57 |
> | VarCon | CIFAR-100 | 78.29 |
> | PaCo | CIFAR-100 | 79.10 |
> | GPaCo | CIFAR-100 | 80.30 |
> | **VarCon+PaCo** | CIFAR-100 | **80.57** |
> | **VarCon+GPaCo** | CIFAR-100 | **81.29** |
>
> Due to the considerable computational overhead of PaCo/GPaCo (particularly PaCo), and given time constraints, we conducted experiments for SupCon, VarCon, and VarCon+PaCo/GPaCo combinations, while using reported results from original papers for standalone PaCo/GPaCo. The results demonstrate that VarCon integrates effectively with PaCo/GPaCo, achieving new state-of-the-art performance: 79.94% on ImageNet and 81.29% on CIFAR-100. We leave comprehensive evaluation on truly long-tailed datasets (CIFAR-100-LT, ImageNet-LT) as future work and would be happy to include these results in the revised manuscript.
>
> > ### Regarding novelty (W3)
>
> We respectfully disagree with the characterization that VarCon builds upon SimVAE. While both use variational inference tools, they address fundamentally different problems with distinct objectives and mechanisms:
>
> **Table 2:** Key differences between SimVAE and VarCon.
>
> | **Aspect** | **SimVAE** | **VarCon** |
> |------------|------------|------------|
> | **Research Goal** | Unified generative theory for SSL | High-performance supervised learning |
> | **Learning Paradigm** | Self-supervised (no labels) | Fully supervised (uses labels) |
> | **Variational Bound** | Data likelihood: $\log p(x\|y)$ | Embedding likelihood: $\log p(z\|r)$ |
> | **Loss Components** | Reconstruction + KL + Prior matching | KL divergence + Log-posterior only |
> | **Reconstruction Term** | Essential (prevents collapse) | None (representation-focused) |
> | **Core Mechanism** | Generative modeling to preserve style | Confidence-adaptive temperature $\tau_2(z)$ |
> | **Architecture** | Encoder + Decoder | Encoder only |
>
> As shown in the table, SimVAE is a self-supervised generative framework that models data generation, while VarCon is a novel supervised contrastive learning method that optimizes embedding distributions. The different research goals, learning paradigms, and mathematical formulations establish them as independent contributions addressing distinct problems in representation learning.
>
>
> > ### Limitation discussion
>
> We thank the reviewer for this suggestion. VarCon reduces computational complexity from $O(B^2)$ to $O(B \cdot C)$ and uses less GPU memory than SupCon (see response to Reviewer gzPQ Q1). Overconfidence is mitigated by our adaptive temperature mechanism (Section 3.2). Calibration under distribution shift is validated through strong transfer learning (Section 4.3, Table 4) and few-shot results (Section 4.2, Table 2). We will include the computational analysis in the revised manuscript and have already discussed broader limitations in Appendix A.1.

---

> > ### Comment · Reviewer_5QVM · 2025-08-04
> > **Official Comment by Reviewer 5QVM**
> >
> > Thank you for the clear and well-organized rebuttal. Your clarifications on the roles of KL, NLL, adaptive temperature, and representation structure addressed my concerns. The distinction from SimVAE is now clearer, and the complementary nature of PaCo/GPaCo is well explained. Based on this, I will raise my score to 4.

---

> > > ### Author Response · Authors · 2025-08-04
> > > **Thank you for your review and recognition**
> > >
> > > Dear reviewer 5QVM,
> > > We sincerely thank you for raising your score as recognition of our work, and we are glad that our response and results have addressed your concerns. We really appreciate your time in reviewing our paper and rebuttals, and your insightful review that helps make our work more comprehensive.

---

### Note · Authors · 2025-08-14

We sincerely thank all reviewers for their invaluable time and effort. We are encouraged by Reviewer 5QVM's recognition of our grounded variational inference and broad empirical gains, Reviewer s4KC's acknowledgment of VarCon's theoretical soundness and superior performance, Reviewer gzPQ's affirmation of our rigorous probabilistic foundation and thorough evaluations, and Reviewer W5fH's emphasis on our innovative confidence-adaptive temperature mechanism and clarity suggestions. We were encouraged that our detailed rebuttals and constructive discussions resulted in consistent positive support from s4KC and gzPQ, and prompted increases in the ratings from gzPQ, 5QVM and W5fH. We appreciate reviewers' suggestions and provide responses to their valuable feedback:

**Novelty and Originality**: We clarified key differences from SimVAE (5QVM) and analyzed our confidence-adaptive temperature mechanism (s4KC), demonstrating dynamic adaptation to sample difficulty. Unlike conventional regularization, our mechanism provides bidirectional adaptation—sharper targets with stronger gradients for hard samples, smoother targets with gentler gradients for easy ones—with implications for generative models and LLM post-training.

**Technical Soundness**: We demonstrated through ablations that KL and NLL terms are not only mathematically necessary for ELBO validity but also empirically essential. Even without adaptive temperature, VarCon outperforms SOTA baseline SupCon, while the adaptive mechanism further enhances results (5QVM, s4KC).

**Evaluation and Method Comparison**: We provided comparisons with PaCo/GPaCo (5QVM) and X-CLR (gzPQ). PaCo/GPaCo's capability in addressing class imbalance integrates synergistically with VarCon, further enabling SOTA performance. For multi-modal X-CLR, VarCon outperforms using only visual data while learning semantic relationships. We focused on single-modality baselines but included these discussions to recognize these methods' contributions.

**Empirical Strength and Efficiency**: We demonstrated all improvements over SupCon are statistically significant (W5fH). Our visualizations show VarCon achieves superior embedding quality in 200 epochs vs SupCon's 350 (5QVM, W5fH). VarCon achieves 79.36% ImageNet with reduced complexity and memory, demonstrating robust improvements with greater efficiency.

We are deeply grateful for the reviewers' constructive feedback and insightful discussions that have significantly strengthened our work.

---

### Decision · Program_Chairs · 2025-09-17

**Decision:**

Accept (poster)

**Comment:**

VarCon presents a theoretically grounded and empirically validated framework that reformulates supervised contrastive learning as variational inference over latent class variables. The method introduces a ELBO-based objective and a confidence-adaptive temperature mechanism, yielding consistent improvements over SupCon across classification, few-shot learning, and robustness benchmarks. Reviewers appreciated the formulation, strong empirical results, and comprehensive rebuttal, with multiple scores raised during discussion. While some concerns were raised about novelty and presentation, the authors convincingly clarified distinctions from prior work and demonstrated the method’s scalability, efficiency, and representation quality. This is a technical sound and potentailly impactful contribution to contrastive learning and representation learning. I therefore recommend acceptance with the expectation that the additional results and clarifications that emerged during the discussion are integrated into the final version.